# Synovial microenvironment-influenced mast cells promote the progression of rheumatoid arthritis

Yunxuan Lei[1,6], Xin Guo[1,6], Yanping Luo[1], Xiaoyin Niu[1], Yebin Xi[1], Lianbo Xiao[2,3], Dongyi He[3,4], Yanqin Bian[3], Yong Zhang[1], Li Wang[1], Xiaochun Peng[5] ✉, Zhaojun Wang [1] ✉ & Guangjie Chen [1] ✉

Mast cells are phenotypically and functionally heterogeneous, and their state is possibly controlled by local microenvironment. Therefore, specific analyses are needed to understand whether mast cells function as powerful participants or dispensable bystanders in specific diseases. Here, we show that degranulation of mast cells in inflammatory synovial tissues of patients with rheumatoid arthritis (RA) is induced via MAS-related G protein-coupled receptor X2 (MRGPRX2), and the expression of MHC class II and costimulatory molecules on mast cells are upregulated. Collagen-induced arthritis mice treated with a combination of anti-IL-17A and cromolyn sodium, a mast cell membrane stabilizer, show significantly reduced clinical severity and decreased bone erosion. The findings of the present study suggest that synovial microenvironment-influenced mast cells contribute to disease progression and may provide a further mast cell-targeting therapy for RA.

Rheumatoid arthritis (RA) is a chronic systemic autoimmune disease that affects approximately 1% of the world's population[1,2]. The peak age for incidence of RA is 50–60 years old, and its frequency in women is 2–3 times of that in men[3,4]. The pathological characteristics of RA include chronic inflammation of the synovial membranes and destruction of bone and cartilage[5]. T cells, B cells, macrophages, mast cells, and other immune cells interact with fibroblasts to promote and maintain local inflammation[6]. Mast cells are innate immune cells derived from the hematopoietic system and widely distributed in tissues directly exposed to the external environment[7]. Once activated, mast cells degranulate to release large amounts of preformed mediators and initiate de novo synthesis of other mediators, including proteases, growth factors, cytokines, and chemokines[8]. Although increased mast cell infiltration in the synovium has been reported in RA[9], it remains unclear whether mast cells act as active participants or just innocent bystanders.

Mast cell activation can be triggered by various classes of receptors[10]. Although mast cells have been considered as primary effectors involved in allergy and asthma through FcεRI, high-affinity IgE receptors, they are also associated with non-allergic conditions, including autoimmune diseases and cancer[11,12]. Increasing evidence shows that MAS-related G protein-coupled receptor X2 (MRGPRX2) can lead to mast cell activation under many non-IgE-mediated conditions[13–15]. Tryptase and chymase-expressing mast cells (MC$_{TC}$), such as skin and synovial mast cells, are the main cells that express MRGPRX2 in peripheral tissues[16]. The ligand-binding pockets of MRGPRX2 are shallow and plastic; thus, the ligands are very broad and include neuropeptides and antimicrobial peptides[17,18]. Upon stimulation, MRGPRX2 causes a

[1]Department of Immunology and Microbiology, Shanghai Jiao Tong University School of Medicine, Shanghai Institute of Immunology, Shanghai, China. [2]Department of Joint Surgery, Guanghua Hospital Affiliated to Shanghai University of Traditional Chinese Medicine, Shanghai, China. [3]Institute of Arthritis Research in Integrative Medicine, Shanghai Academy of Traditional Chinese Medicine, Shanghai, China. [4]Department of Rheumatology, Guanghua Hospital Affiliated to Shanghai University of Traditional Chinese Medicine, Shanghai, China. [5]Department of Orthopedics, Shanghai Sixth People's Hospital Affiliated to Shanghai Jiao Tong University School of Medicine, Shanghai, China. [6]These authors contributed equally: Yunxuan Lei, Xin Guo. ✉e-mail: dr.xcpeng@shsmu.edu.cn; zjwang@sjtu.edu.cn; chenguangjie@sjtu.edu.cn

cascade of downstream signaling pathways, resulting in mast cell degranulation, which may be relatively mild but more durable than IgE-dependent activation[19].

In addition to being effector cells, mast cells can regulate adaptive immunity[20]. In mast cells, expression of MHC class II (MHC II) and costimulatory molecules is inducible in an inflammatory context[21]. TNF enhances PD-L1 expression and IFN-γ upregulates MHC II expression in mast cells[22,23], which might alter B cell and T cell responses. Mast cells can promote immune cell recruitment to inflammatory sites through a wide range of chemotactic factors[24]. Mast cell-derived chemokines, including CCL2, CCL5, and CXCL10, can recruit various T cell subsets[25]. In addition, IL-33-activated mast cells enhance the release of IL-6 and promote T cell polarization[26,27], suggesting that different mast cell phenotypes may change the interplay between mast cells and T cells.

In this work, we find the synovial microenvironment reshapes mast cells into an activated phenotype in RA patients and reveal that the activation of mast cell lines in RA microenvironment is mediated by MRGPRX2, evaluate the efficacy of therapeutic interventions against mast cells in arthritic mice, indicating the crucial role of activated mast cells in disease progression.

## Results

### The increased frequency of synovial mast cells in patients with RA correlates with disease severity

To understand the role of synovial mast cells in RA pathogenesis, we determined the proportion of mast cells in the inflammatory synovium. Summary of patient characteristics in this study was listed in Supplementary Table 1. The percentage of mast cells in synovial single-cell suspensions and in CD45+ immune cells were significantly increased in RA patients compared with that in osteoarthritis (OA) patients (Fig. 1a, b). The same results were observed by immunofluorescence staining for tryptase, a mast cell-specific proteinase. The total number of tryptase-positive synovial mast cells per field in RA patients was significantly higher than that in OA patients (Fig. 1c, d).

OA and RA are both inflammatory diseases[28]. We wondered why mast cells showed increased aggregation in RA patients than in OA patients. As the developmental lineage of human mast cells is still unclear[29], we first hypothesized that mast cells proliferate in situ in the microenvironment associated with disease. Flow cytometry and immunofluorescence staining data showed that synovial mast cells in RA patients rarely expressed the proliferation marker Ki67 (Fig. 1e, f). Contrastingly, the frequency of mast

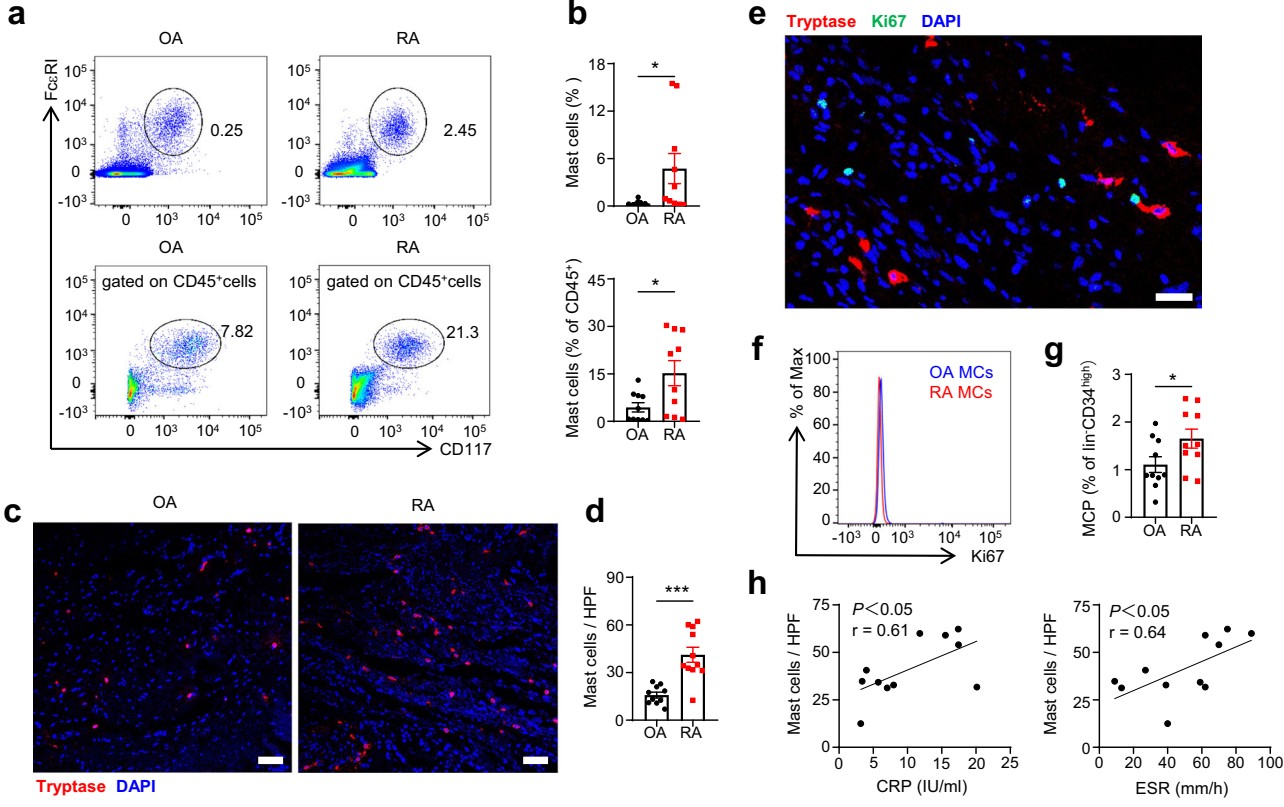

**Fig. 1 | Frequency of synovial mast cells was increased in RA patients and correlated with disease severity. a** Synovial membranes from OA and RA patients were digested into single-cell suspension. Synovial mast cells were gated as CD45+CD117+FcεRI+ cells. Numbers indicated the percentages of mast cells in the gates. **b** Statistical analysis of mast cell percentage in synovial single-cell suspensions and CD45+ immune cells (n = 10 biologically independent samples for each group, P = 0.0328 for percentage in suspensions, P = 0.0196 for percentage in CD45+ cells). **c** Synovial infiltrating mast cells were defined by tryptase immunofluorescence staining. Red, tryptase; Blue, DAPI. Scale bar: 75 μm. **d** Statistical analysis of mast cell numbers per high power field (HPF) in synovial sections (n = 10 biologically independent samples in OA group, n = 11 biologically independent samples in RA group, P = 0.0001). **e** Ki-67 expression on synovial mast cells was detected by immunofluorescence staining. Red, tryptase; Green, Ki-67; Blue, DAPI. Scale bar: 25 μm. Data are representative of three independent experiments. **f** Blue and red histograms in the flow plot depicted synovial mast cells (MCs) from OA and RA patients, respectively. Data are representative of three independent experiments. **g** Frequency of mast cell progenitors (MCP) in the peripheral blood by gating on Lin-CD34hiCD117int/hiFcεRI+ cells (n = 10 biologically independent samples for each group, P = 0.0492). **h** The correlations between synovial mast cell numbers per HPF and ESR or CRP in RA patients were analyzed (n = 11 biologically independent samples, P = 0.0464 mast cells vs CRP, P = 0.0356 mast cells vs ESR). Data are presented as the mean ± SEM and analyzed using two-tailed unpaired t test (**b, d,** and **g**), or two-tailed Pearson's correlation (**h**). *P < 0.05, **P < 0.01, and ***P < 0.001. Source data are provided as a Source Data file.

cell progenitors (MCP), reported as lineage⁻(CD4⁻CD8⁻CD14⁻CD19⁻) CD34^{hi}CD117^{int/hi}FcεRI⁺ cells[30], increased in the peripheral blood of patients with RA (Fig. 1g, Supplementary Fig. 1). Therefore, we suggested that the aggregation of synovial mast cells in RA patients is possibly caused by expansion of mast cell progenitors and their homing towards the synovial membrane. More importantly, the increased number of synovial mast cells was well correlated with the erythrocyte sedimentation rate (ESR) and C-reactive protein (CRP), which represent disease activity in RA[31] (Fig. 1h). Together, these data indicated an increased synovial mast cell burden in patients with RA and a potential role of synovial mast cells in the progression of RA.

## The RA synovial microenvironment reshapes mast cells into an activated phenotype

Mast cells present in different tissues are highly heterogeneous, with various phenotypes that depend on the local microenvironment[32]. To further explore the phenotypes and functions of synovial mast cells, we digested the synovium from patients with OA and RA into single-cell suspensions. Subsequently, mast cells were enriched by CD117

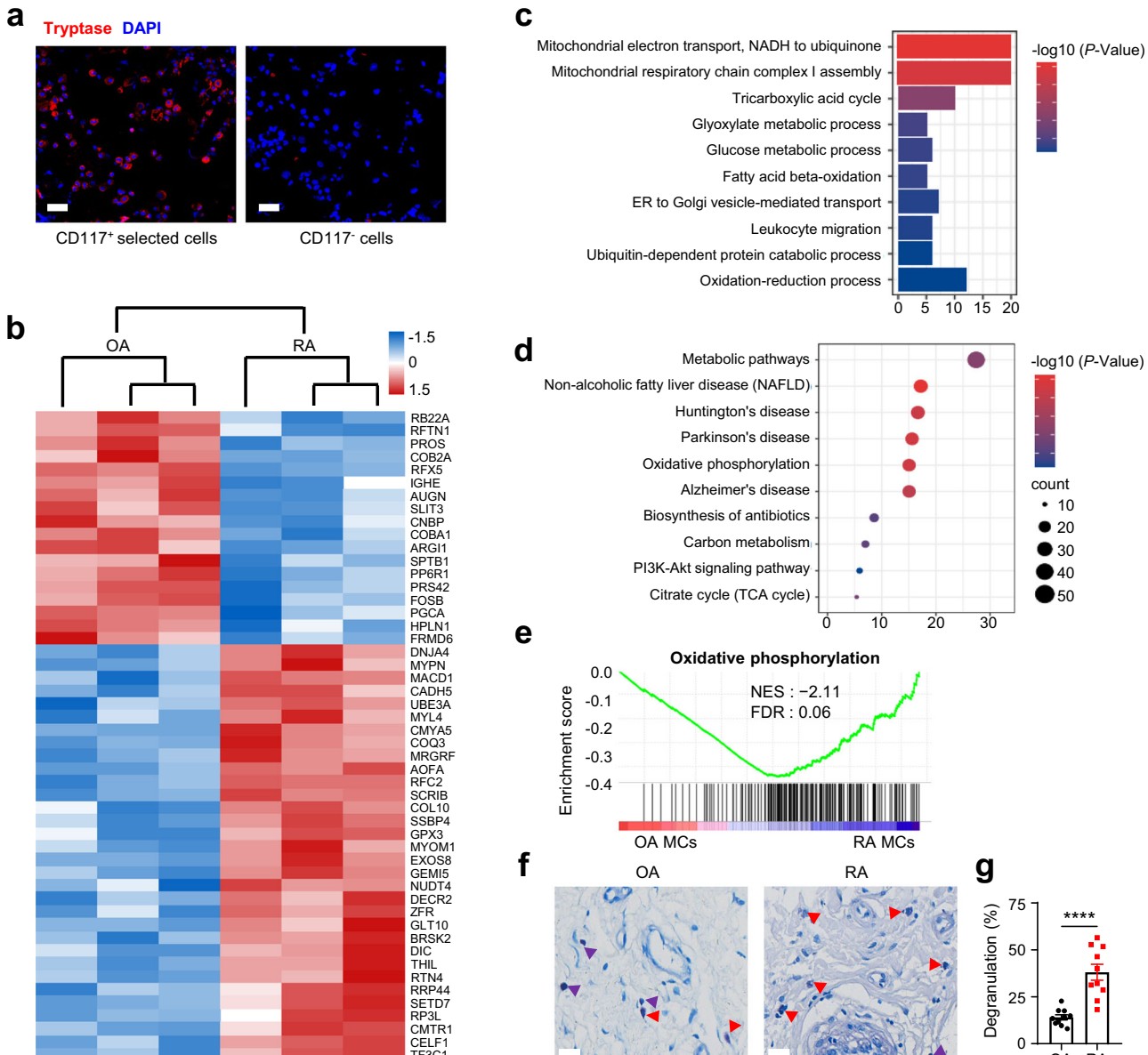

**Fig. 2 | RA synovial microenvironment-induced activation of mast cells.** Synovial mast cells were enriched from digested synovial tissues of OA and RA patients by magnetic bead sorting for CD117⁺ cells and used for label-free mass spectrometry. **a** Mast cell purity was confirmed by tryptase staining of enriched CD117⁺ cells. Red, tryptase; Blue, DAPI. Scale bar: 25 μm. Data are representative of three independent experiments. **b** Heatmap of top 50 differentially expressed proteins between synovial mast cells from OA and RA patients by hierarchical clustering after z-score normalization. Upregulated and downregulated proteins in RA synovial mast cells were depicted in red and blue, respectively. **c, d** GO analysis and KEGG pathway analysis of differentially expressed proteins showed enriched terms including mitochondria metabolism, energy metabolism, and leukocyte migration, along with strong changes in metabolic pathways. **e** Overexpression of oxidative phosphorylation related proteins in synovial mast cells from RA patients by GSEA. **f** Representative images of toluidine blue-stained synovium from OA and RA patients. Resting mast cells and degranulated mast cells were indicated by purple and red markers, respectively. Scale bar: 25 μm. **g** Statistical analysis of mast cell degranulation rate by counting five fields of view (n = 10 biologically independent samples for each group, P < 0.0001). Data are presented as the mean ± SEM and analyzed using two-tailed unpaired t test (**c, d, g**), ****P < 0.0001. The P-values are adjusted in multiple comparisons (FDR < 1% for **c** and **d**). Source data are provided as a Source Data file.

positive magnetic bead sorting. The collected mast cells were used for label-free proteomics after the purity was confirmed by tryptase staining (Fig. 2a).

We identified 8583 unique proteins after matching peptides to proteins in the UniProt database. Principal component analysis (PCA) revealed large variance of protein expression between synovial mast cells from patients with OA and RA (Supplementary Fig. 2a). Compared with that in mast cells from OA patients, expression of 70 proteins in synovial mast cells obtained from RA patients was significantly decreased, and that of 195 proteins was significantly increased (Fig. 2b, Supplementary Fig. 2b). Gene ontology (GO) functional enrichment analysis and signaling pathway analysis based on the Kyoto Encyclopedia of Genes and Genomes (KEGG) database were performed for all 265 differentially expressed proteins. The most enriched GO terms and KEGG pathways were related to mitochondrial respiratory chain complex assembly, electron transport chain, and metabolic process (Fig. 2c, d), indicating an adaptable metabolism change in mast cells. Furthermore, gene set enrichment analysis (GSEA) showed that mast cell metabolism in patients with RA was more inclined to oxidative phosphorylation (Fig. 2e), which was illustrated as a metabolic characteristic of long-term activated mast cells[33]. Therefore, we determined the mast cell activation ratio using toluidine blue staining. The degranulation rate of synovial mast cells in RA patients was higher than that in OA patients (Fig. 2f, g). We validated that mast cells may be shaped into an activated phenotype in the RA microenvironment.

## Activation of mast cells in RA microenvironment is mediated by MRGPRX2

To explore the potential mechanisms behind mast cell activation in patients with RA, we tested the possibility of using synovial fluid from patients with RA (RASF) to imitate the RA microenvironment in vitro. LAD2 cells were labeled with Indo-1, and intracellular calcium flux was detected by flow cytometry upon RASF stimulation (Fig. 3a). Dynamic monitoring for nearly five minutes showed that RASF stimulated increased intracellular calcium flux of mast cells (Fig. 3b). Mast cell degranulation has been reported to be dependent on increased intracellular calcium concentrations[34]. We further stimulated the LAD2 cells with different concentrations of RASF. Determining the upregulation of surface membrane CD63 expression is one of the methods used for quantitative analysis of mast cell degranulation[35]. The increased frequency of CD63-positive mast cells indicated that RASF-induced concentration-dependent degranulation of mast cells (Fig. 3c).

We then screened the downstream signaling pathways related to mast cell activation. Phosphorylation of PLCγ, Akt, and Erk was significantly increased after RASF stimulation (Fig. 3d). As these were reported to be involved in the main signaling pathways of MRGPRX2[36], which explains mast cell activation in many non-IgE-mediated diseases, we speculated that synovial mast cell activation in RA patients was mediated by MRGPRX2. In peripheral tissues, functional MRGPRX2 is selectively expressed on mast cells. We validated significantly higher expression of MRGPRX2 in synovial tissues from RA patients than that from OA patients (Fig. 3e, f). Upregulated levels of cathelicidin LL-37, an antimicrobial peptide, contribute to many autoimmune diseases including systemic lupus erythematosus and psoriasis[37,38]. Furthermore, as a ligand of MRGPRX2[39], the concentration of LL-37 in the synovium of patients with RA was significantly higher than that in patients with OA (Fig. 3g), indicating MRGPRX2-mediated activation of mast cells. LAD2 cells were pretreated with MRGPRX2 siRNA for 48 h, or QWF for 30 min, an inhibitor of MRGPRX2[40,41], before applying RASF. Consistent with our findings, LAD2 cells, where MRGPRX2 was inhibited or knocked down, showed significantly reduced reactivity upon RASF stimulation by low levels of intracellular calcium levels and degranulation rate (Fig. 3h–i, Supplementary Fig. 3a). In addition, RASF-induced phosphorylation of Akt and Erk was significantly weaker

at all time points in MRGPRX2 inhibited or knocked down LAD2 cells (Fig. 3j, Supplementary Fig. 3b). Taken together, MRGPRX2 may be the primary receptor involved in RA synovial microenvironment-induced mast cell activation.

## Adoptive transfer of mast cells promotes disease progression in collagen-induced arthritis (CIA) mice

Considering our in vitro results, we performed mast cell adoptive transfer in a CIA mouse model to assess the effect of mast cell aggregation on disease development. DBA/1 mice were subcutaneously immunized with emulsion of collagen type II (CII) and complete Freund's adjuvant (CFA) at the tail of mice on day 0. The transfer group received tail vein injection of bone marrow derived mast cells (BMMC) on day 7 and day 14, and the control group received equivalent PBS injection at the same time points (Fig. 4a). We found that the transfer group had a statistically higher severity score than the control group, which started to appear from the onset stage (day 35) of the disease (Fig. 4b). At the end of follow-up (day 41), immunofluorescence staining of mouse knee joint synovium showed that the number of mast cells and T cells in the transfer group increased and T cells were infiltrated in the lining layer (Fig. 4c). We also performed mast cell adoptive transfer in serum transfer induced arthritis (STA) mice, another widely used model for RA, and validated that infused mast cells can transfer to arthritic joints (Supplementary Fig. 4).

To further investigate the phenotype of the circulating CD4+ T cells, we harvested spleen and draining lymph nodes at the onset stage (day 35). Healthy mice that received mast cell transfer showed elevated Th1 populations in the spleen and no altered populations in the lymph nodes, nonetheless we observed a significant increase of Th1 and Th17 subsets in lymph nodes from CIA mice that received mast cell transfer (Fig. 4d, e), indicating that mast cells may mediate T cell skewing in inflammatory sites.

In CIA model, CII-specific IgG antibodies (anti-CII IgG) and CII-specific T-cell response play a crucial role for the development of arthritis. Higher concentrations of anti-CII serum IgG and IgG2a in the transfer group indicated a stronger autoimmune humoral response to CII (Fig. 4f). To quantify T cell-mediated immunity, we cultured spleen cells in a medium with 50 μg/mL CII. The proliferation rate of T cells that responded to CII was significantly higher in the transfer group than in the control group (Fig. 4g, h). We collected supernatants from these cells and measured cytokine production. The levels of inflammatory cytokines that mainly derived from spleen T cells, including IFN-γ and IL-17A, showed an obvious increase in the transfer group (Fig. 4i). Overall, these results showed worsening arthritis symptoms in the transfer group, indicating that a higher mast cell burden may promote disease severity in CIA.

## Synovial mast cells in RA microenvironment may regulate T cell response

Our in vivo assays showed an increase in synovial T cell infiltration and T cell mediated immunity in CIA mice after mast cell adoptive transfer. We hypothesized that mast cells interact with T cells to regulate T cell response. First, we analyzed the distribution of mast cells and T cells in the inflamed synovium. Immunofluorescence staining of synovial tissues collected from patients with RA revealed that mast cells appeared frequently where T cell aggregated and formed close contacts with T cells (Fig. 5a). Ectopic lymphoid structures (ELS) refer to ectopic lymphoid aggregates that are mainly formed by B cells and T cells, commonly found at sites of inflammation in autoimmune diseases and cancer[42]. Synovial mast cells even located at the outer boundary of ELS (Fig. 5a). Considering the locations of mast cells and T cells, we performed t-distributed stochastic neighbor embedding (t-SNE) analysis to examine the immunophenotypes of synovial mast cells in patients with RA. Five clusters of synovial mast cells were identified based on SSC and the expression of MHC II, CD40, CD80, and OX40L (Fig. 5b).

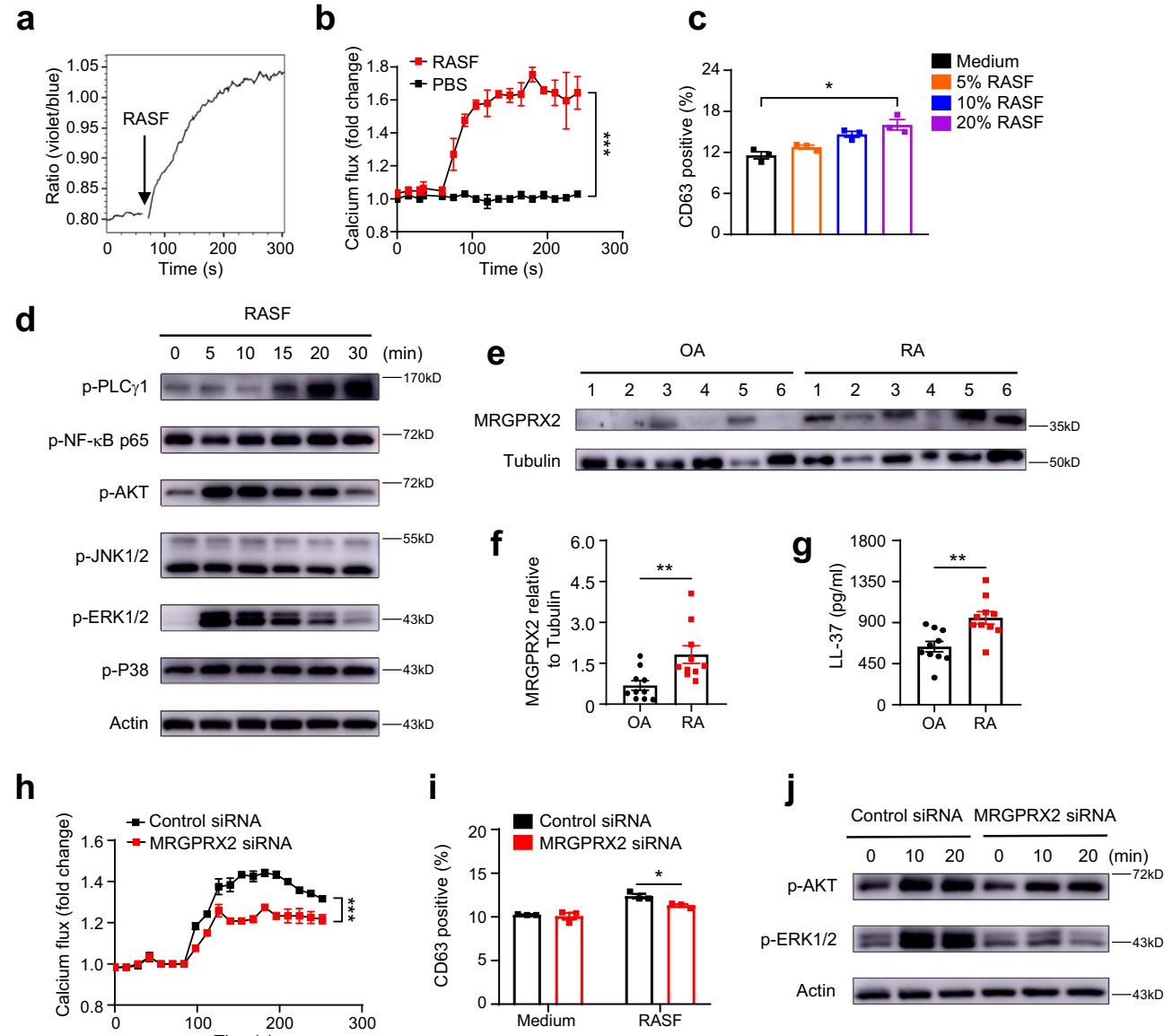

**Fig. 3 | RASF can induce mast cell activation through MRGPRX2.** To imitate RA synovial microenvironment in vitro, LAD2 cells were stimulated with synovial fluid from RA patients (RASF). **a** LAD2 cells were loaded with Indo-1 and applied with RASF. The ratio of $Ca^{2+}$ bound Indo-1 violet to $Ca^{2+}$ free Indo-1 blue was plotted against time and was monitored for nearly five minutes by flow cytometry. The black arrow indicated the time when RASF was added. **b** Intracellular calcium flux (fold change of Indo-1 ratio compared with baseline) of LAD2 cells in response to RASF or PBS was shown ($n = 3$ for each group, pooled from three independent experiments, $P = 0.0002$). **c** Statistical analysis of degranulation rate, indicated by the proportion of CD63⁺ cells in LAD2 after being stimulated with different concentrations of RASF for 20 min ($n = 3$ for each group, pooled from three independent experiments, $P = 0.0139$). **d** Phosphorylated levels of signaling pathways in LAD2 cells treated with 10% RASF for different time intervals. Data are representative of three independent experiments. **e** Western blots of MRGPRX2 and Tubulin in total protein extracts of synovial tissues from OA and RA patients. Data are representative of two independent experiments. **f** Quantification of relative MRGPRX2 expression by densitometry ($n = 10$ biologically independent samples for each group, $P = 0.0068$). **g** Statistical analysis of LL37 concentration in synovial tissues between OA and RA patients ($n = 10$ biologically independent samples for each group, $P = 0.0023$). Intracellular calcium flux (**h**), degranulation rate (**i**), and representative blots (**j**) for LAD2 treated with siRNA against MRGPRX2 or control siRNA ($n = 3$ for each group, pooled from three independent experiments, $P < 0.0001$ for **h,** $P = 0.0362$ for **i**) after RASF stimulation. Data are representative of three independent experiments (**j**). Data are presented as the mean ± SEM and analyzed using two-way ANOVA (**b**, **h**, **i**), Kruskal-Wallis test (**c**), two-tailed Mann Whitney test (**f**), two-tailed unpaired $t$ test (**g**). *$P < 0.05$, **$P < 0.01$, and ***$P < 0.001$. Source data are provided as a Source Data file.

Cells in cluster 2 displayed moderate granularity but high expression of MHC II and costimulatory molecules, which enable mast cells to enhance T cell response[43,44]. We further confirmed MHC II expression by immunofluorescence staining (Supplementary Fig. 5a). In addition, as the expression of these molecules in synovial mast cells of OA patients was relatively lower than that of RA patients (Fig. 5c), we speculated that this unique mast cell phenotype was molded by the RA microenvironment.

Mast cells are known for their ability to release various immune mediators. We stimulated LAD2 cells with RASF collected from four RA patients and detected the expression levels of multiple cytokines and chemokines at different time intervals. Real-time PCR data showed that RASF significantly enhanced expression of *CCL2*, *CCL3*, *CCL4*, *CXCL8*, *IL6*, and *TNF*, which are involved in T cell recruitment and polarization[24] (Fig. 5d). Meanwhile, the expression of genes that encoding regulatory cytokines in mast cells including *IL4*, *IL10*, *IL13*,

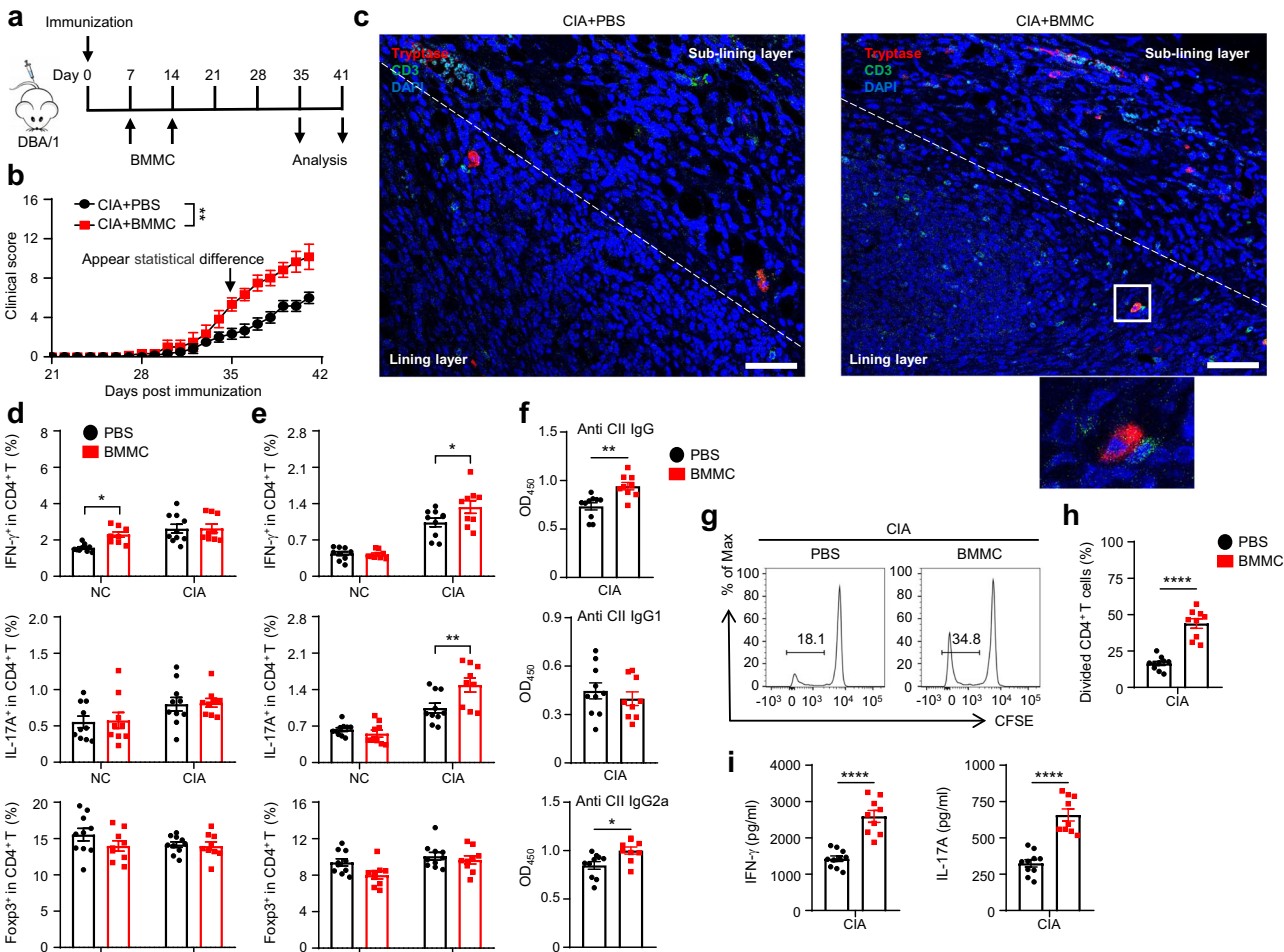

**Fig. 4 | Adoptive transfer of mast cells promoted disease severity and T cell response in CIA mice. a** The schematic diagram of study design: After subcutaneous immunization of CII and CFA at the tail on day 0, 5×10⁶ mast cells were adoptively transferred through tail vein injection on day 7 and day 14. **b** Clinical scores of control group and transfer group (n = 6 mice for each group, pooled from two independent experiments, P = 0.0016). **c** Representative immunofluorescence images of synovial tissues from control group and transfer group. Red, tryptase; Green, CD3; Blue, DAPI. Scale bar: 50 μm. White dotted lines represent the boundary between lining layer and sub-lining layer. The area in the white box is enlarged as an inset. Data are representative of 6 biologically independent samples. **d** Frequencies of CD4⁺IFN-γ⁺ cells, CD4⁺IL-17A⁺ cells, and CD4⁺Foxp3⁺ cells in the spleen detected by flow cytometry (n = 10 mice for PBS, n = 9 mice for BMMC, pooled from two independent experiments, P = 0.0175). **e** Frequencies of CD4⁺IFN-γ⁺ cells, CD4⁺IL-17A⁺ cells, and CD4⁺Foxp3⁺ cells in the draining lymph nodes detected by flow cytometry (n = 10 mice for PBS, n = 9 mice for BMMC, pooled from two independent experiments, *P = 0.0183, **P = 0.0026). **f** Serum levels of anti-CII IgG, IgG1, and IgG2a were quantified by ELISA (n = 10 mice for PBS, n = 9 mice for BMMC, pooled from two independent experiments, **P = 0.0012, *P = 0.0112). **g** Representative flow plots of CFSE-labeled splenocytes gated on CD3⁺CD4⁺ cells after being stimulated with CII for 4 days. **h** Statistical analysis of proliferation rate after CII stimulation (n = 10 mice for PBS, n = 9 mice for BMMC, pooled from two independent experiments, P < 0.001). **i)** Inflammatory cytokines released from splenocytes (n = 10 mice for PBS, n = 9 mice for BMMC, pooled from two independent experiments, P < 0.001). Data are presented as the mean ± SEM and analyzed using two-way ANOVA (**b, d, e**), two tailed unpaired t test (**f, h, i**). *P < 0.05, **P < 0.01, ***P < 0.001, and ****P < 0.0001. Source data are provided as a Source Data file.

*TGFB1* were not affected by RASF (Supplementary Fig. 5b). Overall, these data suggested that mast cells may contribute to T cell response via surface costimulatory molecules and secretion of immunomodulatory factors, thereby promoting the development of RA.

## Combination therapy of cromolyn sodium and anti-IL-17A attenuates CIA

Considering the parallel results of the in vivo and in vitro assays, the crosstalk between mast cells and T cells led us to evaluate whether mast cell blockade can improve the therapeutic effects of biological agents in CIA. IL-17A neutralization approaches have been used in many autoimmune diseases such as psoriasis and RA[45,46]. Unfortunately, some RA patients cannot achieve ideal therapeutic effects or show any response to blockade of IL-17A[47]. Cromolyn sodium is a clinically used mast cell membrane stabilizer that acts by inhibiting the release of chemical mediators from sensitized mast cells[48].

We first examined the efficacy of cromolyn early treatment. Cromolyn was intraperitoneally administered from day 0 (Fig. 6a), and anti-IL-17A was intraperitoneally administered from day 21. Cromolyn and anti-IL-17A were both given every other day. Mice that received combined treatment of cromolyn and anti-IL-17A showed minor wight loss and significantly reduced clinical severity of arthritis relative to mice that only received anti-IL-17A treatment, suggesting that mast cell blockade may potentiate the biological therapy (Fig. 6b, c). Interestingly, we observed that cromolyn alone was sufficient to restrain the progression of CIA relative to mice received vehicle treatment. Histopathological analysis by hematoxylin-eosin (H&E) staining and Safranin O-fast green staining revealed a significant decrease in synovial inflammation and bone destruction in the combined treatment group (Fig. 6d, e). We also found that the concentrations of serum CII-specific antibodies (anti-CII IgG, and IgG2a), as well as serum

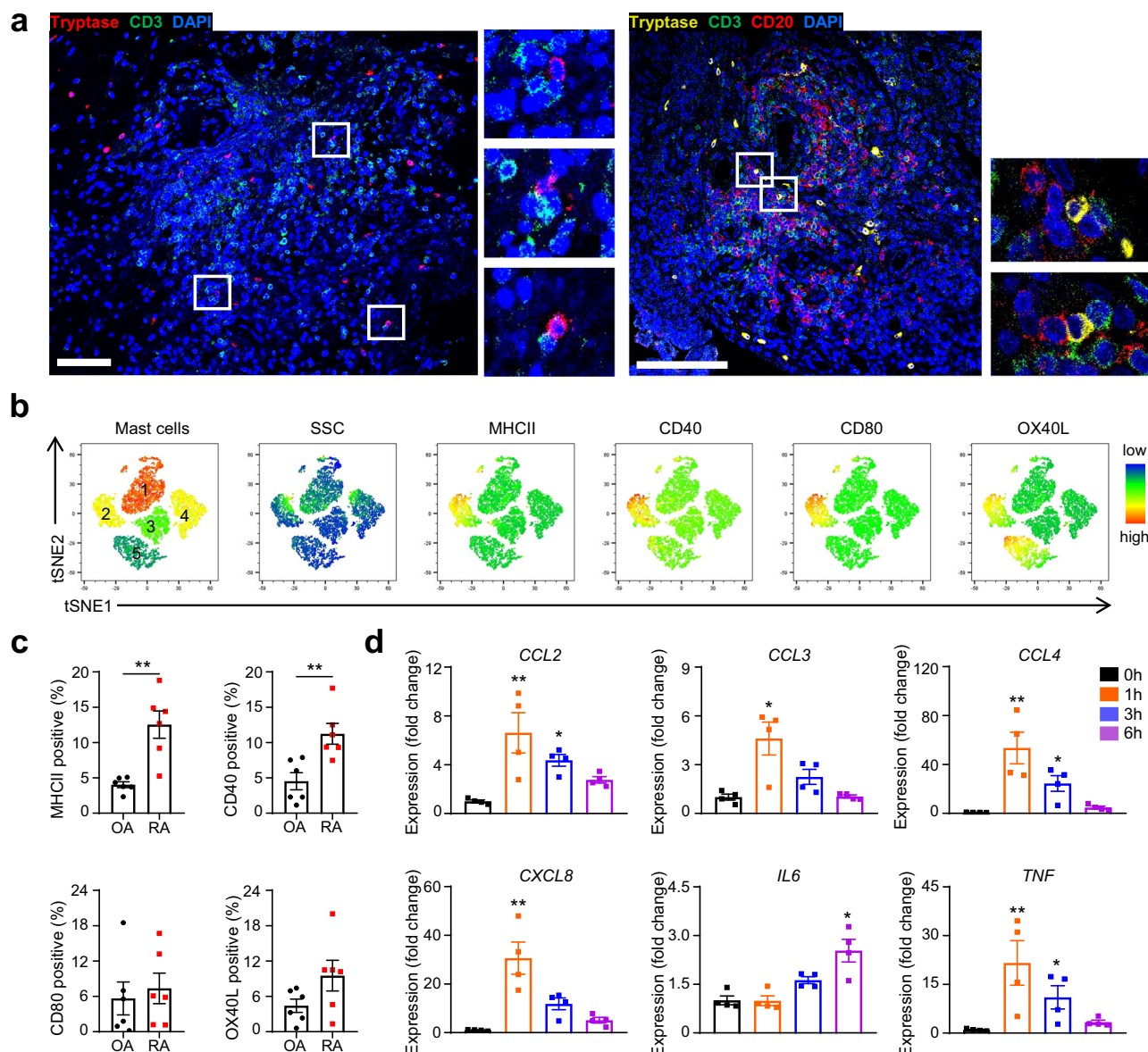

**Fig. 5 | Mast cells under RA microenvironment showed altered phenotypes.**
**a** Representative immunofluorescence images of synovial tissues from RA patients. Left: Red, tryptase; Green, CD3; Blue, DAPI; Scale bar: 75 µm. Right: Yellow, tryptase; Green, CD3; Red, CD20; Blue, DAPI; Scale bar: 100 µm. The area in the white box is enlarged as an inset. Data are representative of 10 biologically independent samples. **b** t-SNE plots of synovial mast cells from four RA patients. **c** Frequencies of MHCII⁺, CD40⁺, CD80⁺, and OX40L⁺ mast cells between OA and RA patients ($n = 6$ biologically independent samples for each group, $P = 0.0022$ for MHCII, $P = 0.0087$ for CD40). **d** Chemokine and cytokine gene expression of LAD2 cells stimulated with 20%RASF for different time intervals were analyzed by qPCR ($n = 4$ biologically independent samples for each group, **$P = 0.007$ and *$P = 0.0225$ for CCL2, $P = 0.0225$ for CCL3, **$P = 0.0025$ and *$P = 0.0347$ for CCL4, $P = 0.0011$ for CXCL8, $P = 0.02$ for IL6, **$P = 0.0054$ and *$P = 0.0311$ for TNF). Data are presented as the mean ± SEM and analyzed using two-tailed Mann Whitney test (**c**), and Kruskal-Wallis test (**d**). *$P < 0.05$, **$P < 0.01$. Source data are provided as a Source Data file.

inflammatory cytokines, including IL-6 and IL-17A, were significantly reduced in the combined treatment groups (Fig. 6f, g). Moreover, spleen T cells from the combined treatment group showed impaired proliferation and produced less IFN-γ and IL-17A in response to CII (Fig. 6h–j). To explore the therapeutic effects of cromolyn in established CIA mice, cromolyn was intraperitoneally administrated from day 21, at the same time point with anti-IL-17A treatment (Supplementary Fig. 6a). In line with early treatment, mice that received combined treatment showed restrained development of CIA but no significant difference relative to mice that received anti-IL-17A treatment (Supplementary Fig. 6b–j).

Taken together, these results suggested that cromolyn can suppress CIA progression and enhance efficacy of biological agents in CIA.

## Discussion
As tissue-resident cells, mast cells are mainly located around nerves and blood vessels, ensuring that they can quickly sense and respond to various challenges[49]. When activated, mast cells tend to infiltrate into inflammatory sites and amplify inflammatory reactions[50]. They monitor and affect many physiological and pathological processes, including immune cell recruitment and tissue remodeling[51]. However, the expansion of mast cells under many inflammatory conditions suggests that their role can transit from immune sentinels to active participants.

Here, we observed a higher mast cell burden in the synovium of patients with RA, accompanied by its strong correlation with the clinical examination index related to disease severity. Mechanisms underlying mast cell expansion in specific diseases are poorly understood. Mouse connective mast cells have been reported to be long-

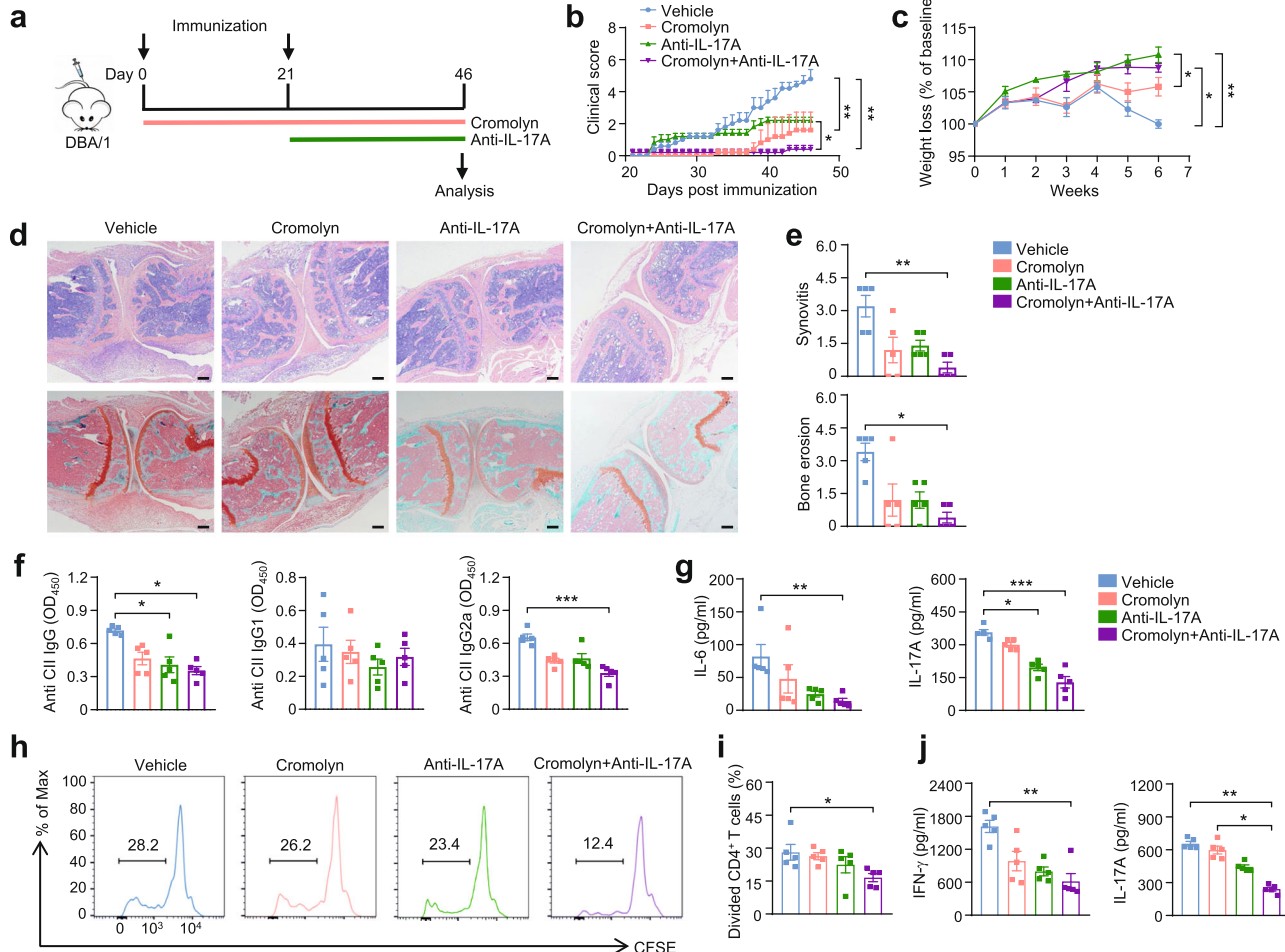

**Fig. 6 | Prophylactic applications of cromolyn sodium and anti-IL-17A attenuated CIA. a** The schematic diagram of study design: All mice received initial immunization on day 0 and booster immunization on day 21. After being randomly divided into 4 groups, mice were treated through intraperitoneal injection with the vehicle, cromolyn (25 mg/kg), IL-17A antibody (150 μg each time), or the combination of the cromolyn and IL-17A antibody. Cromolyn and anti-IL-17A treatment was started from day 0 and day 21, respectively, and were both given every other day. **b** Arthritis scores were measured every day after booster immunization ($n = 5$ mice for each group, $P = 0.0049$ Vehicle vs Cromolyn, $P = 0.0016$ Vehicle vs Cromolyn +Anti-IL-17A, $P = 0.0437$ Anti-IL-17A vs Cromolyn+Anti-IL-17A). **c** Weight loss of each group showed as a percentage change observed once a week ($n = 5$ mice for each group, $P = 0.0013$ Vehicle vs Anti-IL-17A, $P = 0.0165$ Vehicle vs Cromolyn+Anti-IL-17A, $P = 0.0262$ Cromolyn vs Anti-IL-17A). **d** Representative images of H&E and Safranin O-fast green staining of knee joints. Scale bar: 200 μm. **e** Statistical analysis of synovitis and bone erosion ($n = 5$ mice for each group, **$P = 0.0069$, *$P = 0.0119$). **f** Serum levels of anti-CII IgG, IgG1, and IgG2a were quantified by ELISA ($n = 5$ mice for each group, $P = 0.0327$ for IgG Vehicle vs Anti-IL-17A, $P = 0.0166$ for IgG Vehicle vs Cromolyn+Anti-IL-17A, $P = 0.0009$ for IgG2a). **g** Serum levels of IL-6 and IL-17A were determined by ELISA ($n = 5$ mice for each group, $P = 0.0067$ for IL-6, *$P = 0.0234$ and ***$P = 0.0009$ for IL-17A). **h** Representative flow plots of CFSE-labeled splenocytes gated on CD3+CD4+ cells after stimulated with CII for 4 days. **i** Statistical analysis of proliferation rate after stimulation ($n = 5$ mice for each group, $P = 0.0243$). **j** Inflammatory cytokine release from splenocytes after being stimulated with CII ($n = 5$ mice for each group, $P = 0.0034$ for IFN-γ, *$P = 0.0166$ and **$P = 0.0014$ for IL-17A). Data are representative of two independent experiments (**b-j**). Data are presented as the mean ± SEM and analyzed using two-way ANOVA (**b, c**), Kruskal-Wallis test (**e-g**, and **j**), and one way ANOVA (**i**). *$P < 0.05$, **$P < 0.01$, and ***$P < 0.001$. Source data are provided as a Source Data file.

term tissue-resident cells with self-renewal ability in steady state[52]. Human mast cell progenitors are now considered to originate in the bone marrow, then translocate into the peripheral blood and enter peripheral tissue sites[29]. However, a variety of immune cells proliferate in situ[53]. It has been reported that in situ proliferation of Ki67+ MC population was observed in type 2 allergic inflammation[54,55]. Our data showed synovial mast cells in RA patients may not in a proliferative state because of lacking Ki67 expression. In addition, the proportion of mast cell precursors in the peripheral blood of RA patients increased. Considering that aberrant myelopoiesis existed and augmented inflammation in other autoimmune diseases[56], we concluded that synovial mast cells of RA patients are more likely to be renewed and supplemented by bone marrow hematopoiesis under inflammatory conditions.

Cellular metabolic shifts are essential for the activation of many immune cells[57]. Recent studies showed that abnormal energy metabolism of immune cells leads to imbalance of immune homeostasis and destruction of immune tolerance[58], which play a role in the pathogenesis of RA. To satisfy different energy demands, mast cells dynamically modify metabolic activity in the resting and active stages[59]. We performed label-free proteomic analysis of synovial mast cells from patients with OA and RA. The results identified enhanced expression of proteins involved in the oxidative phosphorylation and mitochondria metabolism, which are crucial for degranulation. We hypothesized that these metabolic changes may result in a more active phenotype in the synovial mast cells of patients with RA. This hypothesis was further confirmed by toluidine blue staining of the synovium.

Degranulation is a hallmark of mast cell activation. Plausible pathways of mast cell activation are abundant given the various receptor expression on mast cells[50]. But the primary trigger for the activation of synovial mast cells in RA still remains speculative.

MRGPRX2 was first identified in sensory neurons of the dorsal root ganglia. $MC_{TC}$ are the predominant MRGPRX2 positive cells in peripheral tissues. MRGPRX2 is expressed on the LAD2 cells, whereas the expression on other mast cell lines such as HMC-1 is very low. MRGPRX2 mediates mast cell activation in many non-IgE diseases[60]. In ulcerative colitis, increased expression of adrenomedullin in epithelial cells and fibroblasts leads to mast cell activation via MRGPRX2[36]. In our study, we found significantly higher expression of MRGPRX2 in synovial tissues from RA patients than that from OA patients. Meanwhile, we observed a higher expression of LL37, one of MRGPRX2 ligands, in synovial tissues of RA patients. Moreover, due to LAD2 cells in which MRGPRX2 was knocked down or inhibited showed reduced response to RASF, but not complete resistance, we cannot exclude the possibility that, in the in vivo setting, mast cell activation is mediated by MRGPRX2 and other potential mechanisms. At least these findings indicate that MRGPRX2 partly underlies RASF-induced mast cell activation and may be an additional target for RA therapy.

It has been extensively debated about the role of mast cells in experimental arthritis models, mainly because of controversial results obtained in different mouse strains. Mast cell deficient $Kit^{W/W-v}$ and $Kit^{W-sh}$ mice caused by mutations affecting KIT have been widely used in experiments. These mice showed resistance to arthritis or full development of arthritis[61,62], which may attribute to the depletion of other hematopoietic and non-hematopoietic cells expressing KIT. Recent studies have reported mast cell knockout mice that do not rely on KIT mutations. Mcpt5-Cre-iDTR mice and red mast cell basophil (RMB) mice showed significantly reduced disease severity in CIA[63], especially when mast cells were depleted during the preclinical phase[64]. However, mucosal mast cells are not affected in Mcpt5-Cre-iDTR mice and basophils are depleted in RMB mice. In addition, most of the genetically modified strains of mice are on a C57BL/6 background, which is generally regarded to be resistant to CIA modeling. Therefore, although using these mice can expand the research methods for mast cells, the experimental results should be treated with caution.

Mast cells are evolutionarily conserved;[65] however, they form a unique cell lineage that is distinct from other hematopoietic cells[66]. The developmental origin and local microenvironment, especially the disease microenvironment, may promote the diversity of the phenotypes and functions of mast cells[67]. In recent researches, human mast cells have been reported to upregulate surface expression of mature MHC II and costimulatory molecules induced by IFN-γ treatment and during immune responses[68]. Moreover, mast cells can serve as non-professional or atypical antigen-presenting cells (APCs) for T cells under certain circumstances[69]. Even without antigen, mast cells can stimulate human CD4+ T cells to induce strong T cell proliferation and activation when T cell receptors are triggered[70]. Consistent with these observations, our data defined a special mast cell population in RA patients distinct from that in OA patients with significantly higher expression of MHC II and costimulatory molecules, which laid the basis for mast cell participation in CD4+ T cell response. Moreover, we found that RASF significantly increased the gene expression of various chemokines and cytokines in mast cells. CCL2, CCL3, and CCL4 are known chemotactic factors for different T cell subsets[71]. IL-6 and TNF promote T cell polarization in an inflammatory environment[72]. Taken together, mast cells under RA microenvironment may enhance T cell response.

Multi-center randomized controlled clinical trials have shown that for the early treatment of RA, the remission of clinical symptoms after combination therapy was improved compared to single-drug treatment[73,74]. There is a development trend to explore drug combination therapies. Based on our results, we proposed the combined therapy using cromolyn and anti-IL-17A in CIA mice can greatly impair CIA development and decrease CII-specific T cell recall response. It is important to note that although cromolyn can inhibit mast cell degranulation and enhance the efficiency of biological agents in CIA, intraperitoneal administration of cromolyn may also inhibit degranulation of other cells, such as basophils[75].

This study has several limitations. First, we found that RASF can directly induce mast cell activation through MRGPRX2, but which ligands other than LL-37 in RASF mediated this reaction still needs further investigation. Second, our studies used a limited number of mice to estimate the long-term therapeutic effects of cromolyn on arthritis mice, additional experiments are needed to support our observations and explore other administration methods. Third, although cromolyn is a clinical used medication, sufficient explorations of the benefits and potential side effects of cromolyn treatment in RA should be assessed before the potential clinical practice.

Conclusively, this study provided important insights into the role of mast cells in RA. It demonstrated that synovial mast cells of RA patients were activated by the local microenvironment through MRGPRX2. Synovial mast cells may contribute to disease progression by degranulation releasing proinflammatory factors or via promoting CD4+ T cell response. Combination of cromolyn and anti-IL-17A led to decreased clinical scores and alleviated bone destruction in CIA mouse, indicating drugs that target mast cells can provide a new treatment strategy for RA.

## Methods

### Ethics statement
Our research complies with all relevant ethical regulations. The study involving human samples was approved by the Ethics Committee of Shanghai Sixth People's Hospital Affiliated to Shanghai Jiao Tong University School of Medicine (approval code: 2021-013), and written informed consent was obtained prior to participation. All animal care procedures were performed according to the National Institute of Health Guide for the Care and Use of Laboratory Animals. Animal experiments were conducted in accordance with the guidelines and approval of the Ethical Committee of the Shanghai Jiao Tong University School of Medicine (approval code: A-2019-064).

### Human samples
Fresh synovial tissues and peripheral blood were retrospectively obtained from patients with OA and RA undergoing artificial joint replacement surgery. Patients with other autoimmune diseases, infectious diseases, or cancer were excluded. Specimens were stored in tissue storage solution (Miltenyi Biotec) and immediately transported after surgery. Serum was collected by centrifugation, and peripheral blood mononuclear cells (PBMCs) were isolated using density gradient medium (StemCell).

### Animals
Male DBA/1 J mice and C57BL/6 mice (8- to 10-week-old) were purchased from the Shanghai SLAC Laboratory Animal Center. Male CD45.1+ C57BL/6 mice (8- to 10-week-old) were provided by Prof. Zhaojun Wang, Shanghai Jiao Tong University School of Medicine. All mice were housed under specific pathogen free conditions in the animal care facilities of Shanghai Jiao Tong University School of Medicine, under 12-hour light/dark cycle at 20-24 °C and 45-65% humidity. We used male mice for experiments only, since male mice are more susceptible to arthritis induction. Mice were euthanized by intraperitoneal injection of pentobarbital sodium at indicated analysis time-points.

### Proteomics
CD117-positive synovial mast cells were centrifuged and the cell pellets were frozen in liquid nitrogen. Samples were denatured in 2% SDS buffer containing 50 mM DTT and boiled at 100 °C. The extracted proteins were alkylated using iodoacetamide. After precipitation with precooled acetone, the protein precipitates were digested with sequencing-grade modified trypsin (Promega). Tryptic peptides were

collected by centrifugation, purified using C18 ziptips, and eluted with 0.1% TFA in 50-70% acetonitrile. The eluted peptides were lyophilized using a SpeedVac (Thermo Scientific). The iRT peptides (Biognosys) were spiked into the sample prior to analysis, according to the manufacturer's instructions. DDA was performed using an Orbitrap Fusion LUMOS mass spectrometer (Thermo Scientific). The peptide mixture was loaded onto a self-packed analytical PicoFrit column with an integrated spray tip (New Objective) and separated within a 120-min linear gradient. All MS/MS ion spectra were analyzed using PEAKS 10.0 for processing, de novo sequencing, and database searches. The resulting sequences were analyzed using the UniProt human proteomic database. For all experiments, an FDR < 1% at the peptide spectrum match level was considered. Proteins sharing significant peptide evidence were grouped into clusters and further analyzed using Perseus software.

### Human tissue digestion and mast cell isolation
Fat was removed from the synovium and the synovium was cut into small pieces using sterile scissors. The synovium was then incubated in RPMI 1640 medium (Thermo Scientific) containing 1 mg/mL collagenase I (Sigma), and 0.1 mg/mL DNase I (Roche). Enzymatic digestion was carried out for 1 h at 37 °C, after which the suspension was filtered through gauze and centrifuged at 450×g for 10 min. Pellets were washed with MACS buffer (PBS containing 5% FBS and 2 mM EDTA), and then filtered through a 100 μm and 70 μm cell strainer (Jet Biofil) to obtain a single-cell suspension. CD117-positive cells were isolated from synovial single-cell suspensions using a human CD117 microbead kit according to manufacturer's instructions (Miltenyi Biotec, 130-091-332). The sorted cells were used only when their viability was determined by trypan blue exclusion staining and their purity determined by tryptase staining was > 90%.

### Immunofluorescence staining
Tissues were fixed with 4% paraformaldehyde (PFA; Servicebio) overnight at room temperature, embedded in paraffin, and cut to 3-4um thickness. After deparaffinization and rehydration, sections were immersed in citrate antigen retrieval solution (pH 6.0), maintained at a sub-boiling temperature for 8 min, allowed to stand for 8 min, and then followed by another sub-boiling temperature for 4 min. Sections were permeabilized with 0.5% TritonX-100 (Solarbio) for 20 min, blocked with 20% goat serum (Servicebio) for 1 h at room temperature, and incubated with the primary antibodies overnight at 4 °C in a humidified chamber. The sections were then washed and incubated with secondary antibodies for 2 h at room temperature. Finally, the sections were mounted using DAPI fluoromount (Yeasen) and visualized using a Leica SP8 confocal microscope. For LAD2, the cells were resuspended at 20,000 cells per 100 μL PBS and centrifuged in a Shandon Cytospin 4 cytocentrifuge (Thermo Scientific) at 450×g for 5 min. The cells on the slides were fixed with 4% PFA for 10 min at room temperature and subjected to steps similar to those mentioned above. The primary antibodies used were: anti-mast cell tryptase (Abcam, ab2378, [AA1], 1:500), anti-Ki67 (Abcam, ab92742, [EPR3610], 1:250), anti-CD3 (Abcam, ab16669, [SP7], 1:200), anti-HLA-DR (Abcam, ab92511, [EPR3692], 1:250), anti-CD3 (Abcam, ab11089, [CD3-12], 1:100), anti-CD20 (Abcam, ab78237, [EP459Y], 1:100). The secondary antibodies used were: Alexa Fluor488 anti-mouse IgG (CST, 4408,1:1000), Alexa Fluor594 anti-rabbit IgG (CST, 8889,1:1000), Alexa Fluor488 anti-rat IgG (Thermo Scientific, A21208, 1:400), Alexa Fluor647 anti-mouse IgG (CST, 4410,1:1000).

### Cell culture
The human mast cell line, LAD2, was kindly provided by A. Kirshenbaum and D. Metcalfe (National Institute of Allergy and Infectious Diseases, NIAID). LAD2 cells were cultured in StemPro-34 medium supplemented with 1X nutrient supplement (Thermo Scientific), 2 mM

L-glutamine (Thermo Scientific), 100 U/mL penicillin (Thermo Scientific), 100 μg/mL streptomycin (Thermo Scientific), and 100 ng/mL SCF (Peprotech). Half of the medium was replaced weekly and the concentration was maintained at 0.5-1×10⁶ cells/mL[76]. Bone marrow-derived mast cells (BMMCs) were generated from progenitor cells in mouse bone marrow. Bone marrow cells were isolated from 8- to 10-week-old mice and cultured at 1×10⁶ cells/mL in RPMI 1640 medium (Thermo Scientific), supplemented with 10% FBS (Gemini), 2 mM L-glutamine (Thermo Scientific), 100 U/mL penicillin (Thermo Scientific), 100 μg/mL streptomycin (Thermo Scientific), 10 mM HEPES (Thermo Scientific), 1 mM sodium pyruvate (Thermo Scientific), 1X nonessential amino acids (Thermo Scientific), 50 μM 2-mercaptoethanol (Thermo Scientific), 10 ng/mL IL-3 (Peprotech) and 20 ng/mL SCF (Peprotech). The medium was changed twice a week for 4-8 weeks. Cells were utilized for experiments when their purity, determined by surface staining of FcεRI and CD117 using flow cytometry, was > 90%.

### Calcium flux
LAD2 (2×10⁶ cells) in calcium-free medium (HBSS containing 0.5% BSA) were loaded with 4 μM Indo-1 (Thermo Scientific) for 45 min at 37 °C. Cells were centrifugated at 400×g for 5 min and resuspended in loading media (HBSS containing 1 mM calcium, 1 mM magnesium, and 0.5%BSA). The samples were run at a low speed on an LSR Fortessa X20 (BD Biosciences). After collecting baseline data for 60 s, cells were loaded with different stimulations. The change in intracellular calcium flux was calculated as the intensity of Indo-1 (violet)/Indo-1 (blue) fold change after stimulation compared with baseline data.

### Degranulation
LAD2 cells were washed, suspended in advanced Tyrode's solution (Procell), and stimulated with different concentrations of RASF for 20 min. Thereafter, the cells were washed, fixed with 2% paraformaldehyde and stained with APC anti-human CD117 (BioLegend, 313206, [104D2], 1:100) and PE anti-human CD63 (Thermo Scientific, 12-0639-42, [H5C6], 1:100). CD63 surface expression was detected using an LSR Fortessa X20 (BD Biosciences).

### Induction and assessment of CIA
The CIA model was induced by immunization using a 1:1 mixed emulsion of 4 mg/mL complete Freund's adjuvant (Chondrex) and 2 mg/mL type II collagen (Chondrex). Each mouse received 100 μL emulsion by intradermal injection at the base of the tail. The clinical symptoms in each paw were graded as follows: score 0, normal; score 1, mild swelling or redness of single finger joints; score 2, moderate swelling or redness of more than two joints; score 3, swelling of wrist or ankle joints; score 4, severe swelling of the entire ankle or dysfunction of the limb. The scores for the four paws were totaled, with a maximum score of 16.

### BMMC adoptive transfer and cromolyn treatment strategies in CIA model
Mast cell adoptive transfer was performed on day 7 and day 14 after the initial immunization. BMMCs (5×10⁶ cells) were resuspended in 200 μL PBS and intravenously injected into the transfer group. Equivalent PBS administration was used as control. In cromolyn sodium treatment experiments, 21 days after the initial immunization, each mouse received a booster immunization with 100 μL emulsion of incomplete Freund's adjuvant (Chondrex) and 2 mg/mL type II collagen (Chondrex). Mice were randomly divided into four groups. Mice in the cromolyn group were injected with 25 mg/kg cromolyn sodium every other day from day 0 or day 21. Mice in the anti-IL-17A group received an intraperitoneal injection of 150 μg anti-mouse-IL-17A antibody (BioXcell) every other day from day 21. The combined treatment group

received a combination of the aforementioned treatments. Mice in the vehicle group received solvent equivalent to other groups.

## Induction of STA and BMMC adoptive transfer

Serum from adult, arthritic K/BxN mice was a gift from Prof. Zhaojun Wang, Shanghai Jiao Tong University School of Medicine. Recipient mice were injected intraperitoneally with 150 μL K/BxN serum on day 0 and day 2. CD45.1$^+$ BMMCs ($5 \times 10^6$ cells) were resuspended in 200 μL PBS and intravenously injected into the transfer group on day 1. Equivalent PBS administration was used as control. Spleen and ankle joints were harvested on day 8. Ankle joints were separated and incubated in RPMI 1640 medium (Thermo Scientific) containing 1 mg/mL collagenase D (Sigma). Enzymatic digestion was carried out for 1 h at 37 °C, after which the suspension was centrifuged at 450×g for 10 min. Pellets were washed and filtered through a 100 μm and 70 μm cell strainer (Jet Biofil) to obtain a single-cell suspension.

## In vitro T cell response

Mice were euthanized, and single spleen cells were prepared. Spleen cells were then labeled with 1 μM CellTrace CFSE (Thermo Scientific) according to the manufacturer's instructions. Cells in 96-well round-bottom plates were seeded at $0.5 \times 10^6$/well using RPMI 1640 medium (Thermo Scientific), supplemented with 10% FBS (Gemini), 2 mM L-glutamine (Thermo Scientific), 100 U/mL penicillin (Thermo Scientific), 100 μg/mL streptomycin (Thermo Scientific), 10 mM HEPES (Thermo Scientific), and 50 μM 2-mercaptoethanol (Thermo Scientific), in the presence or absence of CII (50 μg/mL). The cells were cultured for 4 days, then stained with BV421 anti-mouse CD3 (BioLegend, 100228, [17A2], 1:100) and APC anti-mouse CD4 (BioLegend, 100411, [GK1.5], 1:100) after collecting the culture supernatant for cytokine measurement.

## Serum CII-specific IgG detection

Mouse serum was collected by centrifugation of the blood sample at 5000×g for 20 min and stored at −80 °C before use. CII-specific IgG, IgG1, and IgG2a levels were determined by ELISA. Briefly, 10 μg/mL CII diluted in PBS was coated overnight on a 96-well flat-bottom high-binding microplate (Corning). The plates were washed with PBST and blocked for 2 h. After washing, the plates were incubated with the diluted serum for 2 h. Different Igs were detected using 1/5000 diluted HRP-conjugated goat-anti-mouse IgG, IgG1, and IgG2a antibodies (Proteintech). The HRP enzyme activity was measured using TMB (Thermo Scientific). Serial dilutions of pooled serum samples collected from CIA mice were used as the standards.

## Western blot

LAD2 cells were cultured in a cytokine-deprived medium for 6 h, then stimulated with 10% RASF for different time intervals. In some experiments, LAD2 cells were preincubated with different concentrations of QWF (Tocris Bioscience) for 30 min, or in advance knocked down MRGPRX2 by siRNA. After RASF stimulation, the cells were centrifuged, washed with PBS, and lysed using RIPA lysis buffer (Beyotime) with protease and phosphatase inhibitor cocktail (Beyotime). Ten micrograms of protein were loaded for SDS-PAGE. Proteins were transferred onto polyvinylidene difluoride (PVDF) membranes (Millipore). Membranes were blocked with 5% milk solution for 1 h. Primary antibodies were incubated overnight at 4 °C, whereas secondary antibodies were incubated for 1 h at room temperature. Signals were detected using western chemiluminescent HRP substrate (Millipore) and analyzed using Image J. Antibodies used in this study from Cell Signaling Technology: rabbit anti-Phospho-PLCγ1 (CST, 14008, [D6M9S], 1:1000), rabbit anti-Phospho-NF-κB p65 (CST, 3033, [93H1], 1:1000), rabbit anti-Phospho-Akt (CST, 4060, [D9E], 1:1000), rabbit

anti-Phospho-JNK (CST, 4668, [81E11], 1:1000), rabbit anti-Phospho-Erk1/2 (CST, 4370, [D13.14.4E], 1:1000), rabbit anti-Phospho-p38 (CST, 4511, [D3F9], 1:1000), rabbit anti-β-Actin (CST, 4970, [13E5], 1:1000), rabbit anti-α-Tubulin (CST, 2144, 1:1000), and HRP-linked anti-rabbit-IgG (CST, 7074, 1:2000). Antibody used in this study from Abcam: Anti-GPCR MRGX2 (Abcam, ab237047, 1:1000).

## Flow cytometry

Cells ($1 \times 10^6$) were stained with the fixable viability dye eFluor780 (Thermo Scientific) for 30 min at 4 °C, and then stained with an antibody cocktail in FACS buffer (PBS containing 2% FBS and 5 mM EDTA) for 30 min at 4 °C. Intracellular staining was performed using fixation/permeabilization kit (BD Biosciences, 554714) or Foxp3/transcription factor staining buffer set (Thermo Scientific, 00-5523-00), according to the manufacturer's instructions. Initially, for intracellular cytokine staining, cells were restimulated for 5 h with a cell stimulation cocktail plus protein transport inhibitors (Thermo Scientific). The LSR Fortessa X20 or LSR Fortessa flow cytometer (BD Biosciences) and FlowJo were used for detection and data analysis. Antibodies used in this study: FITC anti-human CD45 (BioLegend, 368508, [2D1], 1:100), APC anti-human CD117 (BioLegend, 313206, [104D2], 1:100), PE anti-human FcεRI (BioLegend, 334610, [AER-37], 1:50), PerCP/Cy5.5 anti-human Ki67 (BD Biosciences, 561284, [B56], 1:100), PerCP/Cy5.5 anti-human CD34 (BioLegend, 343521, [581], 1:100), PE anti-human CD63 (Thermo Scientific, 12-0639-42, [H5C6], 1:100), FITC anti-human HLA-DR, DP, DQ (BioLegend, 361705, [Tü39], 1:100), PE/Cy7 anti-human CD40 (BioLegend, 313011, [HB14], 1:100), PerCP/Cy5.5 anti-human CD80 (BioLegend, 305231, [2D10], 1:100), PE anti-human OX40L (BioLegend, 326307, [11C3.1], 1:100), BV421 anti-mouse CD3 (BioLegend, 100228, [17A2], 1:100), FITC anti-mouse CD4 (Thermo Scientific, 11-0041-85, [GK1.5], 1:100), PE/Cy7 anti-mouse IFN-γ (Thermo Scientific, 25-7311-82, [XMG1.2], 1:100), PE anti-mouse IL-17 (BD Biosciences, 559502, [TC11-18H10], 1:100), PE anti-mouse Foxp3 (Thermo Scientific, 12-5773-82, [FJK-16s], 1:50), APC anti-mouse CD4 (BioLegend, 100411, [GK1.5], 1:100), PE-Cyanine7 anti-mouse CD45.1 (Thermo Scientific, 25-0453-82, [A20], 1:100), FITC anti-mouse CD45.2 (Thermo Scientific, 11-0454-82, [104], 1:100), APC anti-mouse CD117 (Thermo Scientific, 17-1171-83, [2B8], 1:100), PE anti-mouse FcεRI (BioLegend, 134308, [MAR-1], 1:50).

## Real-time PCR

Total RNA was extracted using RNAiso Plus (Takara Bio), and cDNA was generated using Hifair III 1st Strand cDNA Synthesis SuperMix for qPCR (Yeasen) and amplified using Hieff qPCR SYBR Green Master Mix (Yeasen) in the presence of specific primer pairs by ViiA 7 Real-time PCR system (Applied Biosystems). The expression of each gene was normalized to TUBB and then to the control. All primer sequences used in this study are listed in Supplementary Table 2.

## ELISA

Cytokine production was determined using mouse IFN-γ, IL-6, and IL-17 ELISA kit (Thermo Scientific, 88-7314-88, 88-7064-88, 88-7371-88), as well as human tryptase and LL-37 ELISA kit (Jianglaibio, JL18959, JL19559), according to the manufacturer's instructions.

## Histopathology

The ankle joints of the mice were fixed using 4% PFA at room temperature overnight and decalcified with 10% EDTA (diluted in PBS) for 4 weeks. The joints were then embedded in paraffin and 4-5 μm serial sections were prepared. Slices were stained with H&E or Safranin O-fast green. Histopathological changes were graded on a scale of 0-4: score 0, normal; 1, hyperplasia of the synovial layer; mild infiltration of leukocytes into the joint; 2, pannus formation; 3, destruction of cartilage; and 4, destruction of bone and extensive infiltrates.

## Transfection

LAD2 cells were washed with antibiotic-free medium and suspended at a concentration of $1\times10^6$ cells/mL. Cells were plated (0.5 mL per well) and then transfected with 100 nM MRGPRX2 siRNA or control siRNA (Ribobio, SIGS0000554-4) using lipo3000 (Thermo Scientific). Knockdown efficiency was detected by western blot 48 h later.

## Statistical analysis

Data analysis was performed using GraphPad Prism and presented as the mean ± SEM. Data normality was examined by the D'Agostino-Pearson test. Unless otherwise stated, comparisons between two conditions with a normal distribution were performed using two-tailed unpaired student's *t*-test and one-way ANOVA was used for comparisons between multiple groups. Mann Whitney test and Kruskal-Wallis test was used for data with skewed distributions between two or multiple groups, respectively. For scoring of arthritis over time, we used two-way ANOVA. Correlations between parameters were assessed using linear regression analysis and the Pearson correlation analysis as appropriate. Statistical significance is indicated as follows: $* P < 0.05$, $** P < 0.01$, $*** P < 0.001$, and $**** P < 0.0001$.

## Reporting summary

Further information on research design is available in the Nature Portfolio Reporting Summary linked to this article.

## Data availability

All data associated with this study are present in the paper and the Supplementary Information or the corresponding author on request. The proteomics data used in this study are available in the Pride database under accession code PXD038584 (http://www.ebi.ac.uk/pride). Source data are provided with this paper.

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

## Acknowledgements

This work was supported by grants from the National Natural Science Foundation of China (81771731, awarded to G.C., 81871269, awarded to X.N.) and Research Project of Shanghai Municipal Commission of Health and Family Planning (202140095, awarded to X.N.). We would like to thank Prof. Tao Zhang for helping obtaining LAD2 cells. We thank Dr. Jia

Jiang for collecting clinical data. We thank Dr. Qi Li and Dr. Xinyue Du for providing experimental materials. We thank Prof. Bing Su and Prof. Zhaoyuan Liu for their insightful suggestions. We thank Proteomics Platform of Core Facility of Basic Medical Sciences, Shanghai Jiao Tong University School of Medicine (SJTU-SM) for performing proteomics and mass spectrometry work. We also thank Hiplot team (https://hiplot.org) for providing valuable tools for proteomics data analysis and visualization.

## Author contributions

G.C. generated the initial ideas and proposed the hypotheses. G.C., Z.W., and X.P. supervised the study. Y.-X.L., X.G., and G.C. conceived and designed the study. Y.-X.L. and X.G. carried out most of the experiments. Y.-P.L. helped with CIA experiments. X.N. and Y.X. provided technical support and suggestions for the project. Y.Z. and L.W. provided technical support. L.X., D.H., and Y.B. assisted with human synovial tissue related experimental design. X.P. provided the human samples. G.C. and Y.-X.L. performed the data analysis, interpretation of the results and wrote the manuscript. All authors read and approved the final manuscript.

## Competing interests

The authors declare no competing interests.
