## [Peer Review File · Nature Communications]

Synovial microenvironment-influenced mast cells promote the progression of rheumatoid arthritisREVIEWER COMMENTS

Reviewer #1 (Remarks to the Author):

In this manuscript the role of mast cells in RA joint inflammation is explored in a human setting as well as in an experimental mouse model. The manuscript provides extensive data on local mast cell tissue adaptation, their contribution to the inflammatory environment and their potential targeting in arthritis. The human data comparing OA and RA seem solid, but the experimental data are based on low numbers and I have some questions about the relevance of the design used.

Questions/remarks:

Medication but also disease duration may have strong effects on the local immune environment. A small table with patient characteristics including amongst others disease duration and medication would be helpful to interpret the results. In figure 2e the authors state that the calcium signaling pathways was enriched in RA mast cells but with a FDR of 0.22 this is not significant.

In Figure 3 the authors state that the activation of synovial mast cells in RA patients is mediated by MRGPRX2. However the experiments on MRGPRX2 activation are performed with the mast cell cell line LAD2, providing supporting but indirect evidence. The authors should therefore adapt their conclusion.

Figure 4

Regarding the experimental CIA model in which mast cells are infused. Mast cells mature in the tissues. Do the mast cells induce systemic effects following the transfer, and how is this reflecting the RA setting? Is there any effect of mast cell transfer in a healthy mouse, e.g, on the T cell skewing in the spleen and LN?

Do the infused mast cells reach the joints? This is not really clear from the histology staining. How was the synovial T cell infiltration in the experimental model determined (line 233)?

N=3 mice per group is a too limited number to perform reliable statistics. In the legend it is mentioned that the data are representative for 2 or 3 experiments. The data from the other

experiments should also be shown or pooled with the rest of the data.

Figure 5.

The authors state that mast cells in RA patients are located near the ELS, and especially co-localize with aggregated T cells. Can the authors quantify this?

In literature some findings suggest a regulatory role in RA by the production of IL-10, TGF- β , IL-4 and IL-13. To claim that the mast cells play a pro-inflammatory role, these cytokines should also be measured.

Figure 6

Why was cromolyn administered at the day of initial collagen immunization, while anti-IL17 was administered at the day of booster immunization? Is the immunization directly affected by the cromolyn? To compare efficacy of anti-IL17 and cromolyn, or the combination, they both should be administered at the time of boosting immunization.

Was this experiment repeated?

Discussion

There are conflicting results in literature about the role of mast cells in experimental arthritis models. Some of these findings should be discussed. For example, depletion of mast cells in the early (but not late) phase of CIA has been shown to reduce the severity of arthritis, also indicating a role in the active immunization process.

Reviewer #2 (Remarks to the Author):

In this manuscript the authors have examined the role of mast cells in rheumatoid arthritis. Mast cells express MAS-related G protein-coupled receptor X2 (MRGPRX2) and proteases tryptase and chymase (MCT, MCTC) and have the potential to upregulate MHC Class II and co-stimulatory molecules and hence activate T and B cell responses.

Others have highlighted roles for mast cells in rheumatoid arthritis and linked them to matrix degradation and inflammation and additionally suggested that targeting these cells might be a therapeutic option and therefore it is important to consider novel elements

presented in this current manuscript.

The authors have compared synovial biopsy material from OA and RA patients and convincingly shown increased presence of mast cells in RA patient material. Metabolic and proteomic profiling of these mast cells showed clear differences the RA mast cells exhibiting a more activated phenotype. The authors have furthermore highlighted their presence adjacent to ectopic lymphoid structure, structures commonly associated with autoimmune conditions with t-SNE analysis splitting the mast cells into five groups one of which in particular showed increased expression of MHC Class II, CD40, CD80 and OX40L.

Support for mast cell potential involvement in RA and an attempt to get at mechanisms of involvement was provided by using LAD2 cells as a surrogate mast cells to dissect the involvement of MRGPRX2 in mast cell activation by RA synovial fluid (RASf) as well as the use of a collagen induced mouse model of RA.

While LAD2 cells may have been useful they are nevertheless not wholly representative of mast cells as they express less tryptase and chymase. It is unclear why the authors did not use isolated mast cells eg from skin. As the authors themselves mention in the discussion they used elevated LL-37 levels in RASf as a rationale for looking at MRGPRX2 but there may well be other molecules in RASf which impact on mast cells. It would have been interesting to see what happens when LL-37 is neutralised. While MRGPRX2 siRNA or the MRGPRX2 inhibitor (QWF Supp Fig S2). reduced the effects of RASf on Ca⁺⁺ flux and phosphorylation of early signalling molecules it did not fully inhibit suggesting that there may be other factors in RASf that may work through other mechanisms on mast cells.

Transfer of bone marrow derived mast cells exacerbated collagen induced arthritis in the recipient mice. Control mice did not receive the two injections of cells or any injections at d7 and d14 and therefore cannot be considered as ideal controls.

It was interesting to see in Fig 4 the increased proportional representation in the draining lymph nodes of T cells expressing IFN γ and T cells expressing IL-17. Sometimes you get cells co-expressing these cytokines and it would have been good to know whether these were such cells or were distinct populations.

The use of cromolyn and anti-IL-17 produced interesting effects on the arthritis scores in the animal model and also on T cell responses and IgG2a collagen antibody levels. Cromolyn, however, does not just target mast cell degranulation but also degranulation by other cell

types and this should be acknowledged.

The authors have by mainly provided enough detail for their experiments to be replicated. It would be useful to know whether the patient material was from patients receiving medication for their inflammatory diseases and if so what treatments. Ideally they would be treatment naive but this seems to have been unlikely in this case given the ages of the patients.

There is novelty in the authors' experimental approach and in their presented data but it does not wholly reveal the mast cell as the definitive player in the pathogenesis of RA but it clearly highlights the potential role for this cell type in concert with others to exacerbate disease.

Reviewer #4 (Remarks to the Author):

Review of NCOMMS-23-02217

Synovial microenvironment-influenced mast cells promote the progression of rheumatoid arthritis

General

The submission is well written and describes an interesting issue of rheumatoid arthritis: the role of mast cells in the synovial fluid or environment of RA. The authors applies various methods to support their conclusion that mast cells promote the progression of disease by facilitating T-cell priming and T-cell functions. Although this might not be wrong, the data dose not directly demonstrate this effect, but only shown that T-cell activation (and activation of other immune reactions) is an event that takes place timely parallel to progression of disease.

Moreover, many of the experiments are performed with few replicate and border-line statistical significance. I strongly advice the authors to consult a biostatistician on the statistics. For instance, t-tests or ANOVA are used when normality cannot be demonstrated (due to low number of replicates). The reader would expect robust data, perhaps especially for a Nature Journal. Having said that, I think the authors are on the verge of making

important and interesting findings, but as the role of mast cells in RA already has been subject to many investigation during the past decades, I think the paper would benefit from more data robustness (more replicates, added controls, and a justified statistical analysis plan).

Specific

L7: Motivators?

L8: RA in full first time

L64: The authors did not explicitly explain why MC showed increased aggregation in RA as compared OA.

L104-106, Fig. 2b: It is not clear to how Fig. 2b describes that 70 or 195 proteins were up/down regulated. I don't identify these number in the heat map. There seem to be 18 downregulated proteins. Explain or revise.

L108: should it be 265 and not 256 (70+195)?

L200: The quality of the CD3 staining in image 4c is not extremely good. There is no comment on the Tryptase staining.

L201: "other in vivo data". Revise style. How can you analyse on 35 when according to Fig 4b, the animals were living through day 41?

L204: How do authors explain the effect on antibodies? How do mast cell affect B cell activation?

L211: Revise as "Overall, these results showed

Fig. 4a: The scheme is incomplete. Please note that tissues were analysed on days 35 and 41.

Fig. 4f: What are the background levels of IgGs for non-CII/CFA-treated mice - performed in the same ELISA? Comment in Results or discussion why the differences in IgG2a is so much stronger than for IgG1?

L227: The presented experiments only contains data from three mice and only reach a $p < 0.05$ significance and only using a non-stringent t test. Therefore, the authors should provide the replicate experiments as suppl. data. Moreover, with $n=3$, no normal distribution can be demonstrated for which reason a parametric test like t-test cannot be done without reasoning. A Wilcoxon or Mann Whitney U test would be more appropriate.

L234: Do MCs in your model regulate T-cell infiltration through histamine? Histamine is a

major preformed mediator released by MCs and it strongly increases vascular permeability. L240/Fig5a: How can the reader identify ELS in the images? The IHC with single antibodies cannot demonstrate co-localisation, as the three sections are obviously from different locations. Also from the IF, co-localisation is not obvious as red and green dots are spread throughout the section and do not necessarily cluster together.

L245: MCs do not present antigen, and MHC and antigen have to be presented together to the T cells, in the T-cell receptor complex. Antigen and MHC and costimulatory molecules (e.g. CD80) cannot be presented by different cells – there is not enough space for that. Hence, explain how MCs contribute to T cell activation/priming.

L250-252: Could the authors perform the experiment with synovial fluid from OA patients or from healthy non-inflammatory individuals? It would be good to know if the effect of chemokines is a special feature of RA or inflammatory fluid in general - or even an effect of any synovial fluid components (hyaluronic acid, lubricin, proteinases, and collagenases).

L253: TNF did not show up in the previous test. Comment?

L256: Again: for T-cell priming, antigen and costimulation takes place at an aggregated synapsis. It is not MHCII/CD80 from one cell and antigen from another APC.

L280: There is no "remarkable" difference between IL17 and combined treatment as weight concerns. Revise, comment.

L281: Please comment more specifically on the extent of the difference and of "good condition". Moreover, the y-axis says "Weight loss". It is rather the weight change relative to baseline. Revise.

L287: "Sharply reduced" is non-scientific. Please revise. Moreover, the combined treatment does not cause reduction compared to the other treated groups of mice, e.g. IL-17. Hence, the authors cannot say that the combination was beneficial over single treatment.

L296: The scheme suggest that cromolyn is administered from day 0. Is it given every other day or daily?

Fig6b-c: combine in one.

Discussion: 1) What role has collagen-specific IgG in the RA? When it drop upon treatment, what does that mean? 2) What causes degranulation of MC in RA? 3) is degranulation Ig-mediated? 4) Do authors consider FcE or FcG to be more involved? If FcG and IgG, platelet activating factor should be measured (or blocked) for verification. 5) Could MC stabilisers be substituted with other drugs, e.g. antihistamine? 6) The role of MC in RA has been known

for decades, even that of MC stabilisers (pubmed.ncbi.nlm.nih.gov/12396022/). What is new here?

L355: How do the MC in RA and AO differ? Or is it just the amount and the activation level that is different?

L357: This has not been experimentally shown. Revise.

L360: I don't see the causality of cytokine/chemokines and effective participation in T-cell priming? Revise.

L366: Over-enthusiastic. Revise.

L387: How many patients? Description of patients (age, sex, etc.)? Level of OA or RA (score)?

L514: Did you measure IgE? After all, MC are associated with IgE. Please provide data or comment.

Detailed response to issue raised by **Reviewer 1** (paraphrased in blue)

In this manuscript the role of mast cells in RA joint inflammation is explored in a human setting as well as in an experimental mouse model. The manuscript provides extensive data on local mast cell tissue adaptation, their contribution to the inflammatory environment and their potential targeting in arthritis. The human data comparing OA and RA seem solid, but the experimental data are based on low numbers and I have some questions about the relevance of the design used.

We thank for your professional review work on our article. We have added more controls and replicates in experiments. According to your comments, we have included some new experimental data to address your concerns. The detailed responses are listed below.

1. Medication but also disease duration may have strong effects on the local immune environment. A small table with patient characteristics including amongst others disease duration and medication would be helpful to interpret the results. In figure 2e the authors state that the calcium signaling pathways was enriched in RA mast cells but with a FDR of 0.22 this is not significant.

We now attached a table of clinical characteristics of patients with OA and RA, to better illustrate the results. GSEA analysis set $FDR < 0.25$ as a reliable enrichment, but we have removed this figure in the revised manuscript after cautious consideration.

2. In Figure 3 the authors state that the activation of synovial mast cells in RA patients is mediated by MRGRPRX2. However, the experiments on MRGPRX2 activation are performed with the mast cell line LAD2, providing supporting but indirect evidence. The authors should therefore adapt their conclusion.

We agreed with the reviewer and carefully revised related description in the results and discussion.

3. Regarding the experimental CIA model in which mast cells are infused. Mast cells

mature in the tissues. Do the mast cells induce systemic effects following the transfer, and how is this reflecting the RA setting? Is there any effect of mast cell transfer in a healthy mouse, e.g, on the T cell skewing in the spleen and LN?

We thank this reviewer for pointing our drawback in experimental designs. Mast cell adoptive transfer by i.v. injection in mast cell deficient mice have been reported to achieve systemic reconstitution (Am J Pathol. 2005;167(3):835-848; Clin Exp Allergy. 2005;35(1):82-88). Considering that we conducted mast cell transfer in normal mice, we checked the distribution of infused mast cell. We observed infused mast cells in lymph nodes of arthritis mice but not healthy mice (Supplementary Fig 4).

We have now added two more groups to observe systemic effects (Fig 4). The data showed that mast cell transfer in healthy mouse caused increased splenic Th1 subsets but showed no significant effects on T cell subsets in draining lymph nodes. Therefore, mast cell transfer may introduce peripheral T cell skewing, without disturbing specific tissues in healthy mice.

4. Do the infused mast cells reach the joints? This is not really clear from the histology staining.

How was the synovial T cell infiltration in the experimental model determined (line 233)?

It is hard to distinguish infused mast cells from synovium resident mast cells by histology staining in our experimental system. Considering the difficulty of using genetic tools in DBA/1 mice, we elected to perform the mast cell adoptive transfer experiment in C57BL/6 mice (Supplementary Fig 4). We conducted serum transfer induced arthritis mouse model by two intraperitoneal injections of 150 μ l K/BxN serum on day0 and day2, and adoptive transferred CD45.1⁺BMMC on day1. We detected the proportion of CD45.1⁺cells in spleen and ankle joints. Transferred mast cells can be found both in peripheral circulation system and arthritic joints.

5. N=3 mice per group is a too limited number to perform reliable statistics. In the legend it is mentioned that the data are representative for 2 or 3 experiments. The data

from the other experiments should also be shown or pooled with the rest of the data.

We feel sorry for the limited number in this part. We tried several different experimental strategies on a small scale of animals. The data cannot be pool together because of the difference between batches which may caused by operation and mice condition.

According to your suggestion, we have replaced Fig4 in the revised manuscript, adding more controls and replicates.

6. The authors state that mast cells in RA patients are located near the ELS, and especially co-localize with aggregated T cells. Can the authors quantify this?

We found our description of “co-localize” may have caused a misunderstanding. So, this statement has now been modified to “mast cells appeared frequently where T cell aggregated and even on the outer boundary of ELS”. We replaced Figure4C with a clearer staining image to better illustrate the adjacent locations of mast cells and T cells. Quantification of MC–T cell contacts by immunofluorescence may be inappropriate because only the layer of slices can be seen.

7. In literature some findings suggest a regulatory role in RA by the production of IL-10, TGF- β IL-4 and IL-13. To claim that the mast cells play a pro-inflammatory role, these cytokines should also be measured.

As requested, we have detected relative expression levels of these genes in LAD2 cells after RASF stimulation and found no significant difference in different time points (Supplementary Fig. 5b).

8. Why was cromolyn administered at the day of initial collagen immunization, while anti-IL17 was administered at the day of booster immunization? Is the immunization directly affected by the cromolyn? To compare efficacy of anti-IL17 and cromolyn, or the combination, they both should be administered at the time of boosting immunization. Was this experiment repeated?

In arthritis mouse model, treatment that start from initial immunization are consider as prophylactic applications and that start from booster immunization are consider as

therapeutic applications. we chose to conduct cromolyn administration from day0 to explore the efficacy of preventive cromolyn therapy. Also, taken poor absorption and short half-life of cromolyn into consideration, we want to undergo long-term treatment. This experiment was repeated once, with n=5, showing similar trends with Fig6 (see below, Reviewer Fig 1).

To address the concerns of the reviewer, we have now performed cromolyn and anti-IL-17 treatment both from the time of booster immunization and showed data in supplementary information (Supplementary Fig. 6). Collectively, these observations showed good therapeutic effects of cromolyn in CIA mice.

Reviewer Fig 1

9. There are conflicting results in literature about the role of mast cells in experimental arthritis models. Some of these findings should be discussed. For example, depletion of mast cells in the early (but not late) phase of CIA has been shown to reduce the severity of arthritis, also indicating a role in the active immunization process.

We thank the reviewer for the important suggestion. We have added more discussion

on the conflicting results obtained in different transgenic mice or animal models.

Detailed response to issue raised by **Reviewer 2** (paraphrased in blue)

In this manuscript the authors have examined the role of mast cells in rheumatoid arthritis.

Mast cells express MAS-related G protein-coupled receptor X2 (MRGPRX2) and proteases tryptase and chymase (MCT, MCTC) and have the potential to upregulate MHC Class II and co-stimulatory molecules and hence activate T and B cell responses. Others have highlighted roles for mast cells in rheumatoid arthritis and linked them to matrix degradation and inflammation and additionally suggested that targeting these cells might be a therapeutic option and therefore it is important to consider novel elements presented in this current manuscript.

The authors have compared synovial biopsy material from OA and RA patients and convincingly shown increased presence of mast cells in RA patient material. Metabolic and proteomic profiling of these mast cells showed clear differences the RA mast cells exhibiting a more activated phenotype. The authors have furthermore highlighted their presence adjacent to ectopic lymphoid structure, structures commonly associated with autoimmune conditions with t-SNE analysis splitting the mast cells into five groups one of which in particular showed increased expression of MHC Class II, CD40, CD80 and OX40L.

Support for mast cell potential involvement in RA and an attempt to get at mechanisms of involvement was provided by using LAD2 cells as surrogate mast cells to dissect the involvement of MRGPRX2 in mast cell activation by RA synovial fluid (RASf) as well as the use of a collagen induced mouse model of RA.

We thank for your positive comments on our article. According to your suggestions, we have made extensive revisions to our previous draft. The detailed point-by-point responses are listed below.

1. While LAD2 cells may have been useful they are nevertheless not wholly

representative of mast cells as they express less tryptase and chymase. It is unclear why the authors did not use isolated mast cells eg from skin.

LAD2 cells express FcεRI, MRGPRX2, and other surface receptors. Also, LAD2 cells contain abundant granules and degranulate well in response to antigen. However, the major drawback for LAD2 cells is relatively deficient production of protease and cytokine, especially when compared to primary cultures of human mast cells. We used mast cell line LAD2 to complete the experiment mainly considering two aspects. One is that mast cells exhibit high tissue heterogeneity, mast cells isolated from other tissues may have different phenotypes. And the other is that it is hard to obtain sufficient quantities of human mast cells from tissues to conduct these studies.

2. As the authors themselves mention in the discussion they used elevated LL-37 levels in RASF as a rationale for looking at MRGPRX2 but there may well be other molecules in RASF which impact on mast cells. It would have been interesting to see what happens when LL-37 is neutralized.

We thank the reviewer for the important question. Finding specific molecules that activate mast cells through MRGPRX2 in the RA microenvironment may be a challenge because that ligands for MRGPRX2 are quite extensive, including certain drugs, endogenous or exogenous peptides. We here proposed that LL-37 may be a potential factor due to recent reports and our data showing high levels of LL-37 in the synovial membrane of patients with RA, but this would require further research. Additionally, we haven't found LL-37 neutralizing antibody yet. LL-37 antibodies on the market can be applied in multiple tests including WB and IHC, but not functional grade for in vitro neutralization.

3. While MRGPRX2 siRNA or the MRGPRX2 inhibitor (QWF Supp Fig S2) reduced the effects of RASF on Ca²⁺ flux and phosphorylation of early signaling molecules it did not fully inhibit suggesting that there may be other factors in RASF that may work through other mechanisms on mast cells.

We agreed with the reviewer. We have added more discussion about it in the revised

manuscript.

4. Transfer of bone marrow derived mast cells exacerbated collagen induced arthritis in the recipient mice. Control mice did not receive the two injections of cells or any injections at d7 and d14 and therefore cannot be considered as ideal controls.

We feel sorry for unclear statement before. We have changed our description of control mice, which received the two injections of PBS at the same time points as transfer group. Moreover, we added two more control group in this experiment now. NC and CIA group that didn't conduct BMDC transfer all received equivalent PBS injection.

5. It was interesting to see in Fig 4 the increased proportional representation in the draining lymph nodes of T cells expressing IFN γ and T cells expressing IL-17. Sometimes you get cells co-expressing these cytokines and it would have been good to know whether these were such cells or were distinct populations.

The ability of T cells to secrete multiple inflammatory factors may be crucial for their pathogenicity. As requested, we analyzed the percentage of IFN γ ⁺IL-17A⁺ cells in CD4⁺ cells but found no significant differences (see below, Reviewer Fig 2).

Reviewer Fig 2

6. The use of cromolyn and anti-IL-17 produced interesting effects on the arthritis scores in the animal model and also on T cell responses and IgG2a collagen antibody levels. Cromolyn, however, does not just target mast cell degranulation but also degranulation by other cell types and this should be acknowledged.

We thank the reviewer for the valuable question. This have been now acknowledged and discussed in the revised version.

7. The authors have by mainly provided enough detail for their experiments to be replicated. It would be useful to know whether the patient material was from patients receiving medication for their inflammatory diseases and if so what treatments. Ideally, they would be treatment naive but this seems to have been unlikely in this case given the ages of the patients.

We have now provided a table of clinical characteristics of patients with OA and RA in the supplemental information. All patients have been diagnosed as arthritis and treated before. The duration of the disease and clinical drug usage have been displayed in the table.

8. There is novelty in the authors' experimental approach and in their presented data but it does not wholly reveal the mast cell as the definitive player in the pathogenesis of RA but it clearly highlights the potential role for this cell type in concert with others to exacerbate disease.

We agreed with the reviewer. We have replaced “definitive player” with “active participants” in the revised manuscript.

Detailed response to issue raised by **Reviewer 4** (paraphrased in blue)

The submission is well written and describes an interesting issue of rheumatoid arthritis: the role of mast cells in the synovial fluid or environment of RA. The authors apply various methods to support their conclusion that mast cells promote the progression of disease by facilitating T-cell priming and T-cell functions. Although this might not be wrong, the data dose not directly demonstrate this effect, but only shown that T-cell activation (and activation of other immune reactions) is an event that takes place timely parallel to progression of disease.

Moreover, many of the experiments are performed with few replicates and border-line statistical significance. I strongly advice the authors to consult a biostatistician on the statistics. For instance, t-tests or ANOVA are used when normality cannot be demonstrated (due to low number of replicates). The reader would expect robust data,

perhaps especially for a Nature Journal. Having said that, I think the authors are on the verge of making important and interesting findings, but as the role of mast cells in RA already has been subject to many investigations during the past decades, I think the paper would benefit from more data robustness (more replicates, added controls, and a justified statistical analysis plan).

We thank for your valuable comments to improve the quality of our manuscript. According to your suggestions, we have revised the manuscript extensively. We discussed more about how mast cell affected T cell response and carefully modified misleading descriptions. Meanwhile, we have added more replicates and controls in several experiments. All data have been analyzed using appropriate statistical methods. Point-by-point responses are listed below.

1. L7: Motivators?

We have changed “motivators” to “participants”.

2. L8: RA in full first time

This have been corrected.

3. L64: The authors did not explicitly explain why MC showed increased aggregation in RA as compared OA.

We thank the reviewer for the important question. Human mast cells progenitors are now considered to originate in the bone marrow, then translocate from the bone marrow into the peripheral blood and enter peripheral tissue sites after homing to various organs (Theranostics. 2020;10(23):10743-10768). However, mechanisms underlying MC expansion in specific diseases are poorly understood. Now, some articles reported that in situ proliferation of Ki67⁺MC population was observed in type 2 allergic inflammation (J Allergy Clin Immunol. 2022;149(6):2062-2077; Sci Immunol. 2021;6(56):eabb7221). In our cohort, synovial mast cells barely express Ki67, but peripheral mast cell progenitors of RA patients significantly increased. Considering that aberrant myelopoiesis existed and augmented inflammation in other autoimmune

disease (Cell. 2022;185(13):2234-2247.e17), we cautiously speculated that the accumulation of synovial mast cells in RA patients was caused by expansion of mast cell progenitors and their homing towards the synovial membrane. Due to the unclear process of how mast cell progenitor cells homing and settling in local tissues, it is difficult to confirm our hypothesis in current research. Therefore, we discussed this in detail at discussion section.

4. L104-106, Fig. 2b: It is not clear to how Fig. 2b describes that 70 or 195 proteins were up/down regulated. I don't identify these number in the heat map. There seem to be 18 downregulated proteins. Explain or revise.

We apologized that the description in the text and figure legends was misleading. We got a total of 265 differentially expressed proteins and showed heatmap of top 50 differentially expressed proteins in the previous manuscript. We have now provided heatmap of all 265 proteins in the supplementary information (Supplementary Fig. 2b) and changed related descriptions in the revised manuscript.

5. L108: should it be 265 and not 256 (70+195)?

Thanks for your careful checks. This has been corrected.

6. L200: The quality of the CD3 staining in image 4c is not extremely good. There is no comment on the Tryptase staining.

We have replaced Fig 4C with a clearer staining containing an enlarged field of view.

7. L201: "other in vivo data". Revise style. How can you analyze on 35 when according to Fig 4b, the animals were living through day 41?

This have been revised. We feel sorry for our misleading description and scheme illustration. To obtain a comprehensive observation of mast cell adoptive transfer in CIA model, we waited for the peak stage of the disease (day41). To obtain in vivo data at the onset stage of the disease, we collected serum, spleen, and lymph nodes at day35. So, clinical score and in vivo data were collected from different independent

experiments. This have been more accurately described in the revised manuscript.

8. L204: How do authors explain the effect on antibodies? How do mast cells affect B cell activation?

It has been established in the literature that human mast cells promote the activation and differentiation of naïve B cells and induce the production of antibodies (Ann Rheum Dis. 2018;77(12):1773-1781). Also, the production of antibodies requires the synergistic effect of B and T cells. We found significantly increased antigen specific T cells in transfer group, which may help with production of antibodies.

9. L211: Revise as "Overall, these results showed

This have been revised as suggested.

10. Fig. 4a: The scheme is incomplete. Please note that tissues were analyzed on days 35 and 41.

This have been revised as suggested.

11. Fig. 4f: What are the background levels of IgGs for non-CII/CFA-treated mice - performed in the same ELISA? Comment in Results or discussion why the differences in IgG2a is so much stronger than for IgG1?

We used ELISA grade collagen II to quantify anti-collagen antibodies in mouse serum, and the readout of OD₄₅₀ in non-CII treated mice were around 0.04, similar to the negative control. High CII specific IgG autoantibody levels are important for CIA model. More specifically, high levels of anti-type II collagen IgG2a subtype antibodies are required to activate complement, an essential step for inducing arthritis.

12. L227: The presented experiments only contain data from three mice and only reach a $p < 0.05$ significance and only using a non-stringent t test. Therefore, the authors should provide the replicate experiments as suppl. data. Moreover, with $n=3$, no normal distribution can be demonstrated for which reason a parametric test like t-test cannot be

done without reasoning. A Wilcoxon or Mann Whitney U test would be more appropriate.

We thank the reviewer for the valuable suggestion. We have replaced Fig4 in the revised manuscript, adding more controls and replicates. Also, we have made sure all data got a justified statistical analysis in the revised manuscript.

13. L234: Do MCs in your model regulate T-cell infiltration through histamine? Histamine is a major preformed mediator released by MCs and it strongly increases vascular permeability.

As requested, we have detected relative expression levels of *HDC*, histidine decarboxylase, in LAD2 cells after RASF stimulation and found no significant difference in different time points (see below, Reviewer Fig 3).

Reviewer Fig 3

14. L240/Fig5a: How can the reader identify ELS in the images? The IHC with single antibodies cannot demonstrate co-localization, as the three sections are obviously from different locations. Also from the IF, co-localization is not obvious as red and green dots are spread throughout the section and do not necessarily cluster together.

Immunohistochemical staining of serial sections may have caused some misunderstandings. We have removed them and changed Fig5a with high-quality images of immunofluorescence staining.

15. L245: MCs do not present antigen, and MHC and antigen have to be presented together to the T cells, in the T-cell receptor complex. Antigen and MHC and costimulatory molecules (e.g. CD80) cannot be presented by different cells – there is not enough space for that. Hence, explain how MCs contribute to T cell

activation/priming.

In recent researches, mast cells have been reported to serve as antigen-presenting cells (APCs) for T cells under certain circumstances (Blood. 2009;114(24):4979-4988; J Allergy Clin Immunol. 2009;123(1):74-81.e1; J Allergy Clin Immunol. 2018;141(1):311-321.e10). Human mast cells and mouse mast cells showed induced surface expression of mature MHC and costimulatory molecules by IFN γ treatment or during immune responses. Mast cells can take up soluble and particulate antigens, process and present antigens to T cells. In summary, mast cells can indeed function as nonprofessional or atypical APCs. Also, even without antigen, mast cells can stimulate human CD4⁺T cells to induce strong T-cell proliferation and activation in the presence of T-cell receptor triggering using anti-CD3(Eur J Immunol. 2016;46(5):1132-1141). According to these reports, we confirmed the expression of MHC and costimulatory molecules on synovial mast cells, which opens the possibility of mast cell mediated antigen presentation. However, it is currently unclear which self-antigens are presented in the RA microenvironment, more research efforts are needed to fully understand whether mast cells actually provide APC functions in RA circumstance. So, we carefully switched the description “contribute to activation/priming” with “enhance T cell response”.

16. L250-252: Could the authors perform the experiment with synovial fluid from OA patients or from healthy non-inflammatory individuals? It would be good to know if the effect of chemokines is a special feature of RA or inflammatory fluid in general - or even an effect of any synovial fluid components (hyaluronic acid, lubricin, proteinases, and collagenases).

We thank the reviewer for the important question. It is hard to collect synovial fluid from non-RA individuals because healthy donors only secrete small amounts of synovial fluid for joint lubrication and OA is characterized by reduction in lubricating synovial fluid.

17. L253: TNF did not show up in the previous test. Comment?

The central role of proinflammatory cytokines, especially TNF, lead to the successful introduction of TNF-inhibitors in the treatment of RA. Given that mast cells can secrete TNF, we detected the influence of RASF on TNF production by mast cells (Fig 5d).

18. L256: Again: for T-cell priming, antigen and co-stimulation takes place at an aggregated synapsis. It is not MHCII/CD80 from one cell and antigen from another APC.

Same response as Question 15.

19. L280: There is no "remarkable" difference between IL17 and combined treatment as weight concerns. Revise, comment.

According to suggestions from the reviewer, we have employed suitable statistical methods and made extensive corrections to our previous draft.

20. L281: Please comment more specifically on the extent of the difference and of "good condition". Moreover, the y-axis says "Weight loss". It is rather the weight change relative to baseline. Revise.

During the course of collagen induced arthritis, swelling and pain in the affected joint would hold back arthritic mice from taking food and gaining weight. So, the degree of weight loss can indirectly reflect the severity of the disease. We have changed "Weight loss (%)" to "Weight loss (% of baseline)".

21. L287: "Sharply reduced" is non-scientific. Please revise. Moreover, the combined treatment does not cause reduction compared to the other treated groups of mice, e.g. IL-17. Hence, the authors cannot say that the combination was beneficial over single treatment.

This have been revised as suggested. We have modified the statistical methods and descriptive terms.

22. L296: The scheme suggest that cromolyn is administered from day 0. Is it given

every other day or daily?

Cromolyn and anti-IL-17A were both given every other day. We claimed it in the figure legends and methods in the revised version.

23. Fig6b-c: combine in one.

This have been revised as suggested.

24. Discussion: 1) What role has collagen-specific IgG in the RA? When it drop upon treatment, what does that mean? 2) What causes degranulation of MC in RA? 3) is degranulation Ig-mediated? 4) Do authors consider FcεE or FcεG to be more involved? If FcεG and IgG, platelet activating factor should be measured (or blocked) for verification. 5) Could MC stabilizers be substituted with other drugs, e.g. antihistamine? 6) The role of MC in RA has been known for decades, even that of MC stabilizers (pubmed.ncbi.nlm.nih.gov/12396022/). What is new here?

Thanks for your valuable suggestions about the discussion. We have revised the discussion section to summarize current understanding of mast cell function in RA and emphasize several new findings of our paper.

1) Collagen-specific IgG represent collagen-specific B cell response, which is affected by the severity of arthritis. Levels of anti-CII IgG are measured for evaluation of arthritis. Medication treatments that have therapeutic effects on arthritis can always lead to decreased antibody levels, whether targeted B cells or not.

2), 3), and 4) Because of the abundant expression of FcγRs and FcεRs on mast cells, these receptors mediated activation have been discussed in other reports (*Clin Exp Immunol.* 1988;71(2):289-294; *Ann Rheum Dis.* 2015;74(10):1915-1923) by in vitro stimulation experiment. But the primary trigger for the activation of synovial mast cells in RA still remain speculative. MRGPRX2 is a newly discovered receptor on mast cells. Here in our article, by inhibiting or knocking down MRGPRX2 on LAD2 cells, we provided MRGPRX2 as a candidate pathway involved in the activation of synovial mast cells in RA. But which ligands are responsible for MRGPRX2 mediated activation still needs more exploration.

5) For better blocking, mast cell activation inhibitions might be more useful than suppressing mediators that have already been released.

6) Although synovial mast cells have been considered involved in the pathophysiology of synovitis, there still many key processes remain unknown. We tried to answer three questions by our finding. Firstly, why mast cells accumulate in synovium of RA patients? This have been mentioned in response to Question3. Secondly, why synovial mast cells activate? We proposed that mast cells may be activated through MRGPRX2 in RA microenvironment. Lasty, can pharmacologic approaches targeting mast cells serve as therapeutic targets in RA? Our data showed that cromolyn improved efficacy of immune checkpoint inhibitor therapy in CIA mice.

25. L355: How do the MC in RA and OA differ? Or is it just the amount and the activation level that is different?

Other than the amount and activation level, we found synovial mast cells showed different phenotypes between patients with OA and RA. Comprehensive phenotypic clustering analysis may require single cell sequencing. But at least by dimension reduction analysis of our flow data (Fig 5b), we found a unique mast cell subtype in RA patients that showed higher expression of MHC and costimulatory molecules.

26. L357: This has not been experimentally shown. Revise.

This have been revised as suggested.

27. L360: I don't see the causality of cytokine/chemokines and effective participation in T-cell priming? Revise.

This have been revised as suggested.

28. L366: Over-enthusiastic. Revise.

This have been revised as suggested.

29. L387: How many patients? Description of patients (age, sex, etc.)? Level of OA or

RA (score)?

We now added detailed table of characteristics of patients in supplementary information.

30. L514: Did you measure IgE? After all, MC are associated with IgE. Please provide data or comment.

We did not measure IgE. Although anti-CII IgE occur in murine collagen induced arthritis. The presence of such IgE antibodies is insufficient, confirmed by comparable levels of anti-CII IgE antibodies in mice with active CIA and spontaneous remission (*J Immunol.* 1991;147(12):4185-4191). Therefore, generally only IgG isotypes are determined in the CIA model.

REVIEWERS' COMMENTS

Reviewer #1 (Remarks to the Author):

The authors have extensively revised the manuscript and have responded in detail to all questions and concerns. They have added new data, to increase statistical power, but also to provide more insights in the experimental models as well as patient information. The discussion section had also been adapted and expanded. I have no further comments.

Reviewer #2 (Remarks to the Author):

The authors have revised their manuscript in the light of the reviewers' comments and provided a detailed rebuttal indicating the changes they have made which are also highlighted in the resubmitted manuscript.

There remain a few changes which this reviewer suggests.

One revolves around LAD2 cells. Figs 3 and Supplementary Fig 3 have titles stating effects on synovial mast cells when in fact performed on LAD2 cells. The titles should reflect what was actually done.

In Fig 4 h & i There is probably a mistake in the legend. The legend states that splenocytes were stimulated with anti- CD3/CD28 or CII. So in i data was pooled from 2 different stimulations? If so, it is remarkable that there is such consistency and there surely is an error here. With anti-CD3/CD28 much more cytokine would be released than with CII. This is clearly seen in the original manuscript Fig4 where they were plotted separately.

In the Materials and Methods section

lines 490, 491 and 492 Secondary antibodies are listed they should define the target and not just the name of the species and so should be listed as anti-mouse Ig, anti-rabbit Ig and anti-rat Ig

The ELISA section line 618 should be updated to include the other cytokines analysed in response to the reviewers comments- IL-4, IL-10, IL-13 and TGF.

The TGF was TGF beta? Was it active that was being assessed- this should be stated.

Reviewer #4 (Remarks to the Author):

The authors have revised and replied satisfactory to my review and provided a better version of the manuscript.

Detailed response to issue raised by **Reviewer 1** (paraphrased in blue)

The authors have extensively revised the manuscript and have responded in detail to all questions and concerns. They have added new data, to increase statistical power, but also to provide more insights in the experimental models as well as patient information. The discussion section had also been adapted and expanded. I have no further comments.

We thank your insightful comments and suggestions to improve our manuscript.

Detailed response to issue raised by **Reviewer 2** (paraphrased in blue)

The authors have revised their manuscript in the light of the reviewers' comments and provided a detailed rebuttal indicating the changes they have made which are also highlighted in the resubmitted manuscript. There remain a few changes which this reviewer suggests.

1. One revolves around LAD2 cells. Figs 3 and Supplementary Fig 3 have titles stating effects on synovial mast cells when in fact performed on LAD2 cells. The titles should reflect what was actually done.

Thank you for pointing this out. We have changed related descriptions in the revised manuscript.

2. In Fig 4 h & i There is probably a mistake in the legend. The legend states that splenocytes were stimulated with anti- CD3/CD28 or CII. So, in i data was pooled from 2 different stimulations? If so, it is remarkable that there is such consistency and there surely is an error here. With anti-CD3/CD28 much more cytokine would be released than with CII. This is clearly seen in the original manuscript Fig4 where they were plotted separately.

Thanks for your careful checks. We presented both anti-CD3/CD28 and CII stimulated results in the original manuscript. To match with other figures, we now only showed CII stimulation. The mistake in the figure legends have been corrected.

3. In the Materials and Methods section: lines 490, 491 and 492 Secondary antibodies are listed they should define the target and not just the name of the species and so should be listed as anti-mouse Ig, anti-rabbit Ig and anti-rat Ig

This have been revised as suggested.

4. The ELISA section line 618 should be updated to include the other cytokines analysed in response to the reviewers comments- IL-4, IL-10, IL-13 and TGF.

The TGF was TGF beta? Was it active that was being assessed- this should be stated.

To imitate the RA microenvironment, we used RASF, which contain a lot of cytokines already, it is not appropriate to conduct ELISA tests in culture supernatants. So, according to the reviewer's suggestion, we detected the gene expression levels of these cytokines by real-time PCR.

The TGF was TGF beta1. Gene name used in the revised manuscript was *TGFBI*.

Detailed response to issue raised by **Reviewer 4** (paraphrased in blue)

The authors have revised and replied satisfactory to my review and provided a better version of the manuscript.

We thank for your valuable comments to improve the quality of our manuscript.